

# Uncertainty quantification of flood mitigation predictions and implications for decision making.

Koen D. Berends[1,2], Menno W. Straatsma[3], Jord J. Warmink[1], and Suzanne J.M.H. Hulscher[1]

[1]Department of Marine and Fluvial Systems, Twente Water Centre, University of Twente, P.O. Box 217, 7500 AE, Enschede, The Netherlands
[2]Department of River Dynamics and Inlands Shipping, Deltares, Boussinesqweg 1, 2629 HV Delft, The Netherlands
[3]Department of Physical Geography, Faculty of Geosciences, University of Utrecht, Princetonlaan 8, 3584 CS Utrecht, The Netherlands

**Correspondence:** K.D. Berends (k.d.berends@utwente.nl)

**Abstract.** Reduction of water levels during river floods is key in preventing damage and loss of life. Computer models are used to design ways to achieve this and assist in the decision making process. However, the predictions of computer models are inherently uncertain, and it is currently unknown to what extent that uncertainty affects predictions of flood mitigation

strategies. In this study, we quantify the uncertainty of flood mitigation interventions on the Dutch River Waal, based on 39 different sources of uncertainty and twelve intervention designs. The aim of each intervention is to reduce flood water levels. Our objective is to investigate the uncertainty of model predictions of intervention effect and to establish relationships to aid in decision making. We show that the uncertainty of an intervention can be adequately described by the newly introduced 'relative uncertainty' metric, defined as the ratio between the confidence interval and the expected effect. Using this metric, we

show that intervention effect uncertainty behaves like a traditional backwater curve with a constant relative uncertainty value. In general, we observe that uncertainty scales with effect: high flood level decreases have high uncertainty and conversely, small effects are accompanied by small uncertainties. However, different interventions with the same expected effect do not necessarily have the same uncertainty. For example, our results show that the large-scale but relatively ineffective intervention of floodplain smoothing by removing vegetation, has much higher uncertainty compared to alternative options. Finally, we

show that for a defined standard of acceptable uncertainty, interventions need to be over-designed to meet this standard, and by how much. In general, we conclude that the uncertainty of model predictions is not large enough to invalidate model-based decision making, nor small enough to neglect altogether. Instead, uncertainty information can be used to improve intervention design and enrich the decision making process.

## 1 Introduction

The number of people living in areas exposed to river flooding is projected to exceed one billion in 2050 (Jongman et al., 2012). Therefore, it is increasingly important that the river system is designed in such a way that flood risk is minimised. Human intervention in river systems has a long history, to a point that for some rivers human management has become the



dominant factor driving hydrological change (Pinter et al., 2006; Bormann and Pinter, 2017). Today, the decision to change an existing river system (e.g., by leveeing a channel to protect flood-prone areas) is increasingly based on predictions made by computer models. Various software systems can be used to resolve flow conditions accurately, taking into account much of the complexity of rivers — such as terrain geometry, vegetation and hydraulic structures. However, despite increasingly available

data, not all model input or model parameters can be reliably measured or represented. For example, vegetation density and height, which modify vegetation roughness (Baptist et al., 2007; Luhar and Nepf, 2013), are variable both in time and space. This variability is neither captured on the scale of river modelling, nor by the equations that resolve vegetation roughness. Examples such as this introduce uncertainty in the modelling process and in the model output (Oreskes et al., 1994; Walker et al., 2003). Proper understanding and communication of this uncertainty is important both for scientists and decision makers

(Pappenberger and Beven, 2006; Uusitalo et al., 2015).

Predicting the effect of flood risk mitigation measures with models is similarly affected by uncertainty. However, there are two compounding issues particular to intervention design studies. First, the detailed models used in effect studies have a high number of parameters that need to be determined. There is a clear practical need for detailed models given the increasingly complicated design of interventions, which moves from traditional flood prevention (building dikes, or levees) to more holistic

designs. This is in part motivated by the paradoxical 'levee effect' that states that flood control measures do not decrease, but increase flood risk (White, 1945; Baldassarre et al., 2013; Munoz et al., 2018). In some countries, this insight has spurred policy change away from purely flood control and towards alternative options (Pinter et al., 2006; Klijn et al., 2018). In The Netherlands, this has led to designs that focus on increasing conveyance capacity to achieve lower flood levels, while integrating multiple other (ecological, societal) objectives as well (Rijke et al., 2012). Predicting the effects of more intricate alternative

approaches, such as the construction of artificial secondary channels (van Buuren and Warner, 2014), requires detailed models that take into account local geometry, vegetation and other terrain features.

A second compounding issue is that effect studies *ex ante* rely (by definition) on simulations of at least one unmeasured environmental system; namely, the river system altered by the proposed intervention. This complicates using measurements to estimate model accuracy, because there is no way to verify that accuracy for the proposed future state of the river. A further

complication is that the hydraulic conditions, for which flood mitigation measures are designed, tend to have a low period of return, e.g. a one in one hundred year or even one in 1250 year flood (Klijn et al., 2018). Therefore, it is likely that measurements of the current, unaltered state are likewise unavailable. For these reasons, the uncertainty of model output cannot be expressed in terms of accuracy (e.g. standard deviation of model error) without additional untestable assumptions.

Model output uncertainty can alternatively be expressed in terms of the uncertainty of model input and parameters. This

approach, also known as forward uncertainty analysis (Beven et al., 2018), is stochastic: both uncertain model input and parameters, as well as model output are expressed as probability density functions. For most practical applications, neither the deterministic problem (i.e. solving the flow equations) nor the stochastic problem (i.e. resolving the model output probability density functions) are analytically tractable. Monte Carlo simulation (MCS) is a well-known method to solve the stochastic problem numerically (Stefanou, 2009). The disadvantage of MCS is that it comes with a significant, often prohibitive, compu-

tational cost. An alternative, computationally less expensive, approach was proposed by Berends et al. (2018). This approach



reduces the number of post-intervention model runs by using the pre-intervention model as a surrogate in a statistical Bayesian framework, which becomes more efficient with an increasing number of interventions.

To this date, there is little evidence in literature of explicit uncertainty quantification for model predictions of flood mitigation measures, or of other human intervention in river systems. Lack of explicit uncertainty quantification leaves room for what
Pinter (2005) calls "fuzzy math" — free interpretation of model uncertainty — by decision makers. In his example, building permits for floodplains were easily granted on the premise that individual effects are negligible compared to model uncertainty. Since the cumulative effect of all those permits is not negligible (Pinter et al., 2008), this interpretation likely overestimates the effect of uncertainty. An opposite example is given by Mosselman (2018), who reports that large, quantified uncertainties in absolute flood levels are sometimes ignored for intervention effects under the assumption that uncertainties 'cancel out'. This
assumption is incorrect (Berends et al., 2018), but further study is needed to determine how large the uncertainty actually is.

Given the problem that effect studies use complicated models in unmeasured situations, it is likely that there is significant uncertainty involved with model predictions. Yet, lacking uncertainty quantification leads to a free interpretation of the importance of this uncertainty. To approach this problem we present a quantitative, real-world study on the effect of model uncertainty on designing flood mitigation measures. Here we restrict the definition of effect to the initial reduction of flood
water levels by a proposed intervention, without considering secondary effects such as subsequent morphological or ecological system response. Our objective is twofold. First, we quantify the effect of parameter uncertainty on the predicted effect of flood mitigation measures, by implementing twelve different interventions of varying type and intensity in a section of the Dutch River Waal. The second objective is to establish a relationship between the expected reduction of flood levels and the uncertainty, to aid in the decision making process. To this end we will introduce the 'relative uncertainty' measure to facilitate
inter-comparison between different intervention designs.

Our analysis proceeds in three steps. First, we set up the numerical model for the Dutch River Waal and select the uncertainty sources in section 2. Here, we also introduce the relative uncertainty measure. We then present the results of the uncertainty of the modelled water levels and reduction in water levels in section 3.1 and 3.2. In section 3.3 we show how uncertainty may influence decision making. In the discussion (4) we discuss reducing uncertainty in intervention design and the feasibility of
probabilistic analysis for intervention design in practice. A general conclusion on the practical value of uncertainty quantification for decision making in flood mitigation strategies, and specific conclusions related to the objectives, are given in section 5.

## 2 Methods

### 2.1 Study area

The River Waal is selected as the case study for its extensive history of human intervention and good data availability. The Waal is a distributary of the River Rhine and, by discharge, the largest river in the Rhine-Meuse-Scheldt Delta, located in Western Europe. The present-day river has a main channel about 8 meters deep at bank-full discharge, a bank-full width of about 250 m and relatively narrow floodplains. Before the construction of the dikes from 1000 A.D. onward, it was a meandering river with



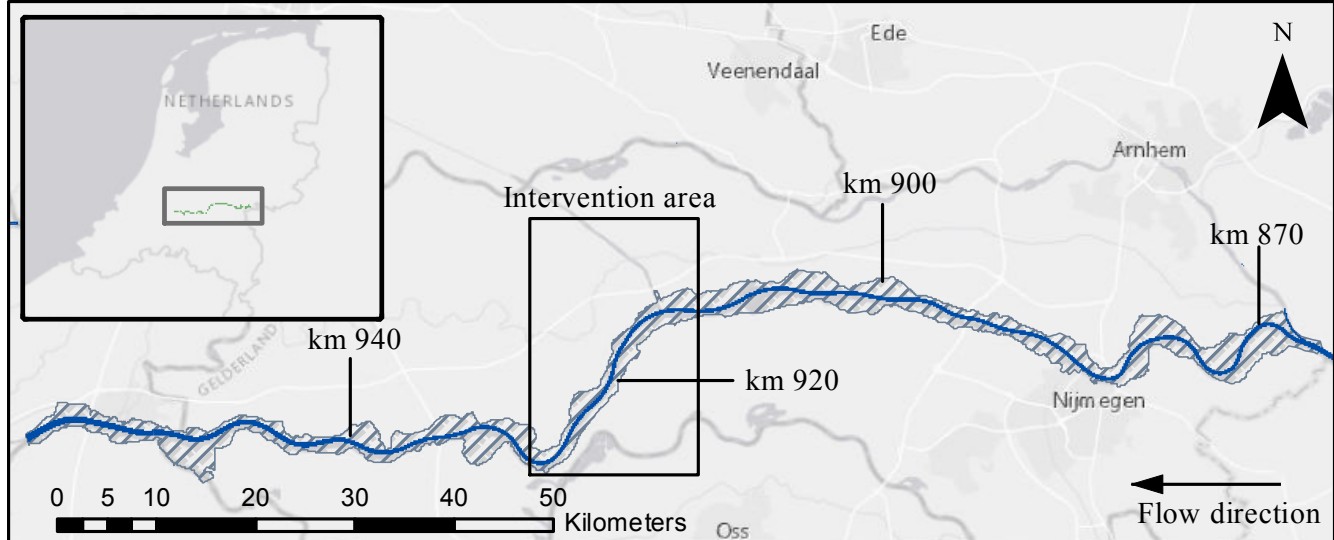

Esri, HERE, Garmin, © OpenStreetMap contributors, and the GIS user community

**Figure 1.** The River Waal, situated entirely in The Netherlands. The study area is a stretch of about 15 km between km 913 and km 928. The hatched area shows the total extent of the model domain, which overlaps with the river and its floodplains bounded by the dikes.

regular over-bank deposition (Hobo et al., 2014). Recorded river engineering works for flood protection as well as as inland navigation dates back to at least the 18th century (e.g., see Velsen, 1749). In the 19th century, the river was normalised in large governmental projects, obtaining its current narrow channel, groynes (also termed *spur dams* or *wing dams*) and relatively straight channel (Sieben, 2009). In present day, the river has a bed slope of approximately $1 \cdot 10^{-4}$ m/m over a stretch of 90

5    km from the upstream bifurcation to the Merwede bifurcation. At the end of the 20th century, motivated by the near-disaster of 1995 (Chbab, 1995), the large scale 'Room for the River' programme was enacted to reduce flood levels (Rijke et al., 2012; Klijn et al., 2018). The largest intervention in the River Waal in this programme was the Nijmegen-Lent dike relocation project, which was predicted to lower the water levels by 27 cm at a cost of approximately 350 million Euro. Most other interventions on the River Waal were projected to lower the water level by less than 10 cm.

10    Our study domain covers the entire Waal River (Figure 1). Distances along the river are measured using the conventional 'Rhine kilometre', which starts from km 0 at the German city of Konstanz. The Waal starts at km 867 at the Pannerden bifurcation and runs through to km 961 where it bifurcates in the Nieuwe Merwede and Boven Merwede rivers. All flood mitigation measures studied will be implemented between km 913 and km 928, depicted by the annotated rectangle in Figure 1.

15    The intervention area is characterised by a relatively straight river stretch with narrow floodplains and the strongly curved *St. Andries* river bend, which provides a known bottleneck during high discharges. Nine towns border directly on the dikes along this 17.5 km stretch. Interventions in this area to increase flood safety, considering the narrow floodplains and the populated surrounding land, present both a technical and a societal problem.




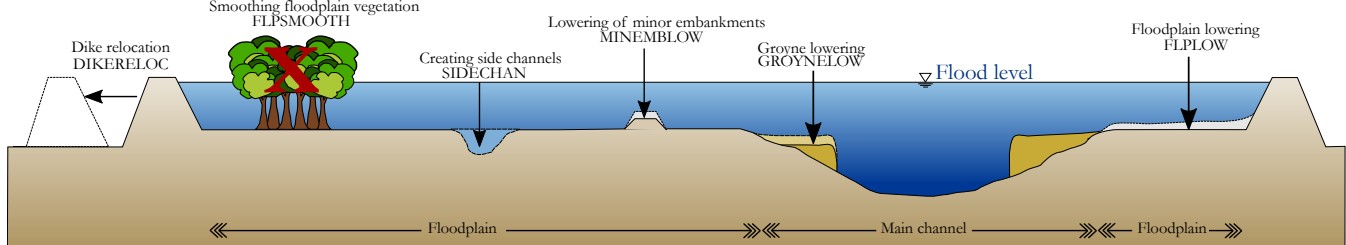

**Figure 2.** A schematic overview of a typical cross-section of the River Waal and the flood mitigation measures studied in this paper. Figure based on Middelkoop et al. (1999).

## 2.2 Hydrodynamic model & interventions

To simulate the hydrodynamic response of the flow to various interventions we use the Delft3D Flexible Mesh modelling system (Kernkamp et al., 2011). Geographical information, such as land cover, bathymetry and hypsometry, as well as information on embankments and weirs, were derived from the Baseline information system (Becker et al., 2014). Our setup uses a two-

dimensional unstructured numerical grid, with between 71,000 and 120,000 active grid cells. The resolution varies from 40m by 15m in the main channel and a gradually increasing resolution in the floodplains to a maximum of about 120m by 120m. In all cases, we simulate a steady upstream discharge of 10.165 $\mathrm{m^3s^{-1}}$ and a constant downstream water level of 3.98 m + NAP, for 72 hours. These conditions are consistent with a return period of 1250 years ($T_{1250}$). Initial conditions were derived from a reference run with all considered stochastic variables (see section 2.3) at their median values. The initial time step was set at

30s.

All interventions are modelled as changes to a reference state of the system. This reference state is the Waal River after all interventions from the Room for the River programme, which approximately corresponds to the 2016 situation. We consider twelve additional system states, each one corresponding with a particular human intervention (Figure 2). In all cases, the interventions were procedurally generated by the RiverScape tool (Straatsma and Kleinhans, 2018). We implemented six different

interventions, detailed below. Each interventions was implemented in a low-intensity and a high-intensity variant. We assume that the intervention is carried out in reality exactly as it was designed. This is known as the *'as designed'* post-intervention state. In reality, there may be a discrepancy between the *as designed* state and the actual (*'as built'*) state.

### 2.2.1 Groyne lowering (GROYNLOW)

Groynes (also known as *wing dams* or *spur dams*) are (stone) structures perpendicular to the flow. During normal conditions,

groynes restrict the effective channel width to promote navigable depths. However, during high flows groynes obstruct flow. Prior to 2016, many groynes in the Waal were already lowered as part of the Room for the River program. The intervention as implemented in this study further reduces the groynes crest-heights. Groynes were lowered to the crest-height corresponding to the water levels with exceedance frequency of 150 days (low intensity) and 363 days (high intensity) per year.





### 2.2.2  Minor embankment lowering (MINEMBLOW)

The River Waal floodplains are compartmentalised by minor embankments, which original purpose is to prevent flooding of the floodplains during minor (summer) floods. This intervention lowers the crests of these embankments to reduce their obstruction during high flow. Low and high intensity lowered the crests to a water level with exceedance frequency of 50 days (low intensity) and 150 days (high intensity) per year.

### 2.2.3  Floodplain lowering (FLPLOW)

Lowering or excavation of the floodplains increases the maximum water volume within the existing bounds of the river corridor, thereby increasing conveyance. In this study, we lower the level of the floodplain without changing existing vegetation or other floodplain configuration to isolate the effect of lowering. Floodplains were lowered to the corresponding water level with exceedance frequency of 20 days per year, with 5% (low intensity) and 99% (high intensity) of the terrestrial floodplain area altered.

### 2.2.4  Floodplain smoothing (FLPSMOOTH)

Vegetation in the floodplain greatly contributes to resistance during high flows. Replacing existing high-friction vegetation by low-friction vegetation mitigates this problem. Here, the existing vegetation was replaced by production meadow (roughness code 1201 in table A1). Smoothing at low intensity affected the top 5% of the obstructing vegetation. At high intensity all vegetation was converted to meadow.

### 2.2.5  Dike relocation (DIKERELOC)

In the River Waal, dikes are the primary defence against flooding. However, they also contribute to flood risk by restricting the river corridor. Dike relocation increases the floodplains and allocates more space to the river. While this is perhaps the best example of combatting problematic constriction of the river corridor, it is also the most invasive — considering human settlement near and on the dikes. At low intensity, concave dike sections of less than 700 m are replaced by straight dikes, whereas at high intensity concave sections of 7000 m are straightened, while creating small polders around existing built-up areas.

### 2.2.6  Side channels (SIDECHAN)

Finally, the construction of secondary (side) channels within the existing corridor both increases the available volume, as decreases vegetation friction. All new channels are assigned "side channel" roughness (Code 105 in Table A1) and a trapezoidal cross sectional shape and slope of 1 to 3. The channels were implemented with widths of 10 m (low intensity) and 100 m (high intensity), and depths of 0.35 m (low intensity) and 3.5 m (high intensity) below the water level with an exceedance frequency of 363 days per year.





**Table 1.** An summarised overview of the stochastic variables. A full overview is given in Table A1

| Type | Number of stochastic variables |
|---|---|
| Classification error | 1 |
| Main channel roughness | 1 |
| Vegetation height ($h_v$) | 14 |
| Vegetation density ($n_v$) | 17 |
| Non-vegetation roughness | 6 |
| Total | 39 |

## 2.3 Uncertainty sources

Modelling real-world rivers on the scale of nearly 100 km necessarily involves various simplifications, discretisations and parameterisations. Sources of uncertainty for this river were identified by Warmink et al. (2011) using expert elicitation. The main sources were (a) boundary conditions and (b) main channel friction and to lesser extent (c) friction by vegetation, (d) geometry and (e) weir and groynes formulations. In this study we follow the design approach taken in the Room for the River program, which assumes the boundary conditions given and stationary for a certain design return rate (which is $T_{1250}$) and therefore not a source of uncertainty. Following Warmink et al. (2013b), we take into account uncertainty in the main channel and classification errors in the land-use maps. Additionally, we take into account uncertainty in vegetation parameters. This leads to a total of 39 stochastic variables (see Table 1).

### 2.3.1 Main channel roughness

The hydraulic roughness of the main channel is chiefly determined by the material of the bed ("grain roughness") and the bed forms ("form roughness"). Various models have been proposed that calculate the friction factor based on the characteristics of the bed (van der Mark, 2009). Here, we adopt the approach of Warmink et al. (2013a) who estimated the values for Nikuradse roughness height $k_n$ [m] for a $T_{1250}$ event by extrapolation using a general extreme values distribution (GEV, Weibull variant) for five roughness models. Given the GEV percentiles, we fitted a lognormal distribution at the $T_{1250}$ return period for each roughness model. These lognormal distributions were combined into a single distribution using equal weight for each roughness model. Figure 3 shows the resulting individual and combined probability functions from the Weibull extrapolation.

In the following we will use the joint cumulative density function of the five roughness models to sample representative roughness height $k_n$. The joint cumulative function is a highly asymmetrical distribution with a mean of approximately 0.58 m and a 95% confidence limits at 0.31 m and 1.0 m. The sampled values of $k_n$ will be used in the hydrodynamic model as input for the White-Colebrook model:

$$C = 18_{10} \log \frac{12h}{k_n} \tag{1}$$





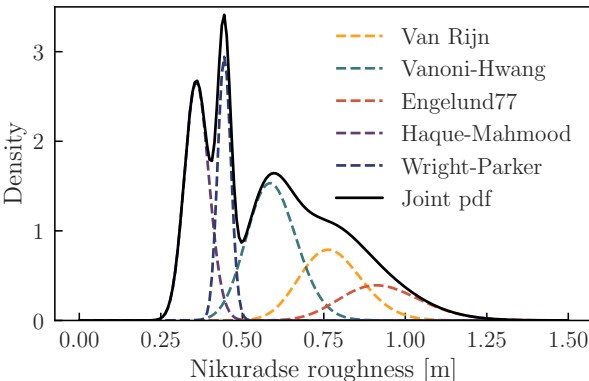

**Figure 3.** The combined probability function of the main bed roughness is based in Weibull extrapolation of Nikuradse roughness height $k_n$ to the $T_{1250}$ return period using five different bed roughness formulas.

with Chézy coefficient $C$ [$\mathrm{m}^{1/2}\mathrm{s}^{-1}$] and water depth $h$ [m]. The sampled $k_n$ values are used to resolve friction for all main channel roughness classes, which cover 95.6% of the main channel area. The other 4.4% are non-erodible layers in the outer river bends which are not stochastically determined in this study.

### 2.3.2 Floodplain roughness

The Baseline database divides the floodplain into vegetation classes, riverbanks and water bodies. Each class is given a specific roughness code called a *trachytope*, which is then coupled to an appropriate roughness formula and parameters specific to that formula. Spatially, the vegetation in the floodplains is discretised by polygons, which are designed either one or a combination of distinct vegetation classes.

For vegetation, the generation of roughness by vegetation is complex, resulting from obstruction by stems and leafs and
turbulent flow through the vegetation and over the canopy. The relationship between vegetation and roughness is extensively studied, leading to various models to compute hydraulic friction from vegetation traits (Nepf, 2012; Vargas-Luna et al., 2015; Shields et al., 2017). These traits often include the vegetation canopy height $h_v$[m], drag coefficient $C_D$ [-], stem density $n_v = m_v D_v$ or vegetation concentration $\lambda = \frac{\pi D_v^2 m_v}{4}$ with number of stems per square meter $m_v$ [$\mathrm{m}^{-2}$] and stem diameter $D_v$ [m] (Klopstra et al., 1997; Stone and Shen, 2002; Baptist et al., 2007; Huthoff et al., 2007; Yang and Choi, 2010; Li et al.,
2015). Warmink et al. (2013b) have shown that the choice for a particular vegetation model has little influence on the outcome uncertainty compared to parameter uncertainty. Therefore, we only used the model proposed by Klopstra et al. (1997), in which vegetation friction is chiefly determined by the parameters $h_v, n_v, C_D$. These parameter values are determined in the following way. For $C_D$ we use fixed, deterministic values based on van Velzen et al. (2003). The height and density parameters ($h_v$ and $n_v$) are assigned probability distributions based on the field campaign of Straatsma and Alkema (2009). Based on 206
field observations, they approximated the variation in vegetation density and height for 25 homogeneous vegetation classes.





Based on their estimate quantiles we fitted lognormal distributions for each class, assuming the vegetation height and density to be independent parameters. In total, the roughness of 66% of the River Waal floodplain is defined by one of the vegetation trachytopes and 2.5% is defined by combination of two vegetation trachytopes.

Six trachytopes are associated with water bodies or riverbanks, accounting for about 27% of the total areal. We model the
friction for these 6 classes with the Manning formula

$$C = h^{1/6} n^{-1} \tag{2}$$

with Manning coefficient $n$ [sm$^{-1/3}$]. The $n$ values for these trachytopes are modelled as triangular probability functions with the minimum, maximum and mean values based on Chow (1959). All parameters of the vegetation parameter distributions are summarised in Table A1. The remaining 4.5% of floodplain area, which is given deterministic roughness values, is mostly
(3.7%) covered by buildings which are excluded from flow computations, revetment and pavement.

### 2.3.3   Classification error matrix

Available vegetation maps are likely to contain 'impurities' (Knotters and Brus, 2012) or classification errors, that have significant impact on model output uncertainty (Warmink et al., 2013a). The probabilities of one class being in reality another is encoded in a confusion matrix (we use Table 1 in Straatsma and Huthoff (2011)). Following this matrix it is found, for example,
that "Willow plantation" was, in all cases, found to be "Softwood forest" instead. An overview of the various vegetation classes is given in the Appendix. Following these probabilities we generated an ensemble of 2000 pre-intervention trachytope maps. Each map was then given a fixed index number. We sample from these maps by picking a number from a discrete uniform distribution and finding the corresponding map by index number. In this way, each map is assigned the same probability.

### 2.4   Quantification of model output uncertainty

### 2.4.1   Estimation method

In this study we focus on prediction of the maximum flood levels (see Figure 2) in the reference state (denoted by $X$), the maximum flood levels in the various interventions (denoted by $Y$) and the difference between these two (denoted by $\Delta H$). Due to the uncertainties in model input discussed in section 2.3, $X$, $Y$ and $\Delta H$ are stochastic. To compute differences between these two (stochastic) states of the river system, we would need to obtain both the model output probability distributions, as
well as the covariances between the two distributions (Berends et al., 2018). This problem is not analytically tractable, but can be solved numerically by Monte Carlo simulation (MCS; Metropolis, 1987; Stefanou, 2009). However, MCS is not practically feasible; although MCS has been applied for estimation of flood level uncertainty before (Warmink et al., 2013b), the added task of performing a separate MCS for each intervention puts a severe strain on computational resources.

Therefore we follow the approach of Berends et al. (2018), schematically depicted in Figure 4, which is computationally
more efficient. We refer to this method as 'correlated output regression analysis' (CORAL). The key feature of CORAL is that the stochastic intervention output $Y$ is not directly solved by MCS, but instead an estimator $\hat{Y}$ is computed using the reference

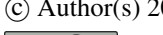


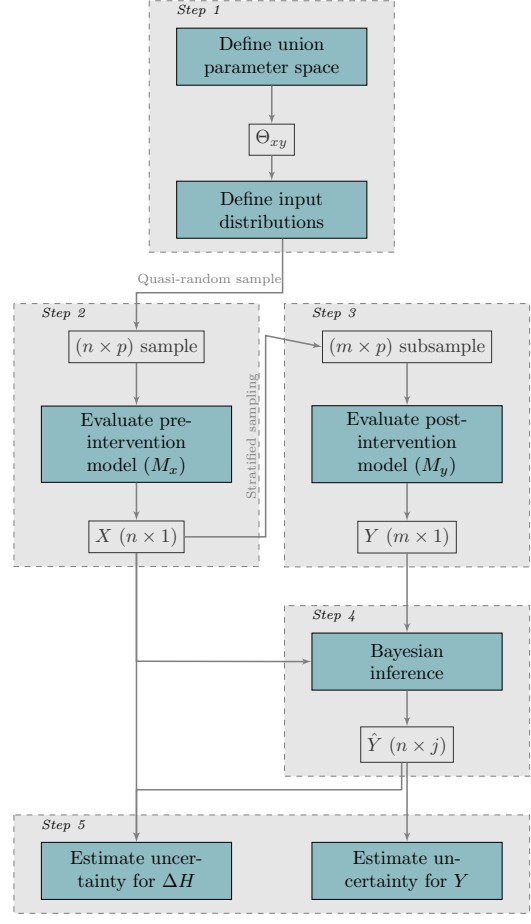

**Figure 4.** Schematic overview of the efficient uncertainty estimation method CORAL for post-intervention model output and impact analysis. In this figure: $n$: MCS sample size; $m$: subsample size; $p$: number of stochastic parameters; $X$: pre-intervention model output; $j$ the number of draws from the posterior predictive model; $Y$: post-intervention model output; $\hat{Y}$: estimated post-intervention model output; $\Delta H$: intervention effect. Figure adapted from (Berends et al., 2018).

state and a correlation model. This correlation model is defined within a Bayesian framework and is probabilistic as well. We use the same linear model as Berends et al. (2018):

$$\hat{Y} = \alpha(X - c) + \beta + \epsilon + c \tag{3}$$

with $\epsilon \sim \mathcal{N}(0, \sigma_\epsilon)$ and transformation constant $c$. The constant is introduced to shift data to the origin, which facilitates faster

5     inference, and is defined as $c = \min(X)$. In the Bayesian framework, the regression parameters $\theta = \{\alpha, \beta, \sigma_\epsilon\}$ are considered stochastic variables themselves and estimated via a Markov Chain Monte Carlo (MCMC) algorithm, trained by a limited subsample of simulations from $Y$. We use a subsample of $m = 20$ for all intervention states. To estimate the distribution of the reference state (step 2) we use a quasi-random MCS sample ($n = 1000$) with the Sobol' low-discrepancy sequence. The linear



model was demonstrated to be applicable to similar intervention studies by Berends et al. (2018), and (visual) inspection of the correlation between the reference state and the intervention models shows this holds for our cases as well.

The main advantage of this approach is that a significant decrease of required computational resources by reducing the number of model evaluations. The total number of hydrodynamic model evaluations was 1240, divided over thirteen model

states (the reference plus 12 interventions). This is a decrease of computational effort of more than 90% compared to direct MCS with all states, which would have totalled 13.000 model evaluations. It is important to state that the consequence of having a probabilistic correlation model, is that the intervention state estimator $\hat{Y}$ is stochastic for each individual draw from the reference state ($X$). Since $X$ is stochastic itself, this can be thought of as the 'uncertainty of the uncertainty' or the estimation uncertainty.

To maintain correlation between the $X$ and $Y$, a union stochastic parameter space is constructed (step 1 in figure 4). Practically, this means that each corresponding member in the reference and intervention model ensembles has identical values for their stochastic variables, with the noticeable exception where those were altered by the intervention. For example, we assume that a tree with a height of 8 m in the reference case still has a height of 8 m after intervention, unless this tree was removed or otherwise affected by the intervention. This is straightforward to implement for parameter values, but requires an additional

step for the classification uncertainty. To be able to draw from the confusion matrix (see section 2.3.3) a large number of trachytope maps was generated for the reference case. To maintain correlation, the same number of maps is generated for each intervention in such a way that every map is identical to the corresponding reference map with the same index, except in areas where the intervention has altered the trachytope. For example, an area which was classified as 'willow plantation' in the reference state will still be classified as 'willow plantation' after the intervention, unless that area has been reclassified as part

of the intervention.

### 2.4.2  Uncertainty metrics

As a data reduction step, we summarise the results using the following metrics based on the water level reduction effect $\Delta H$.

Confidence intervals measure the distance between two percentiles (e.g. the 10% and 90%) of a cumulative density function (cdf). We refer to the confidence intervals of the stochastic variables $X, Y$ and $\Delta H$ as the model uncertainty confidence interval

or MCI. For example, the MCI of $\Delta H$ is defined as:

$$\text{MCI}_{\Delta H} = |\hat{F}_{\Delta H}^{-1}(p_u) - \hat{F}_{\Delta H}^{-1}(p_l)| \tag{4}$$

The MCI of $\hat{Y}$ is derived in the same way. Here, $\hat{F}^{-1}$ is the inverse cumulative density function, also known as the *quantile* or *percent point* function and $p_l$, $p_u$ the lower and upper quantiles. Here, these quantiles are always symmetric around the median. We use a limited subset of quantiles (consistent with the 10%, 20%, 50%, 80% and 90% confidence intervals) to summarise

the data. Additionally, we calculate the exceedance probability $P(\Delta H < z) = \hat{F}_{\Delta H}(z)$.

It is important to note that $\text{MCI}_{\hat{Y}}$ and $\text{MCI}_{\Delta H}$ are stochastic — resulting from the estimation method. Therefore, each value belonging to an arbitrary quantile $p_z$ may be expressed in terms of the expected value $E(\hat{F}_{\Delta H}^{-1}(p_z))$ and variance





$\text{Var}(\hat{F}_{\Delta H}^{-1}(p_z))$. This uncertainty may be referred to as the estimation uncertainty. A more detailed account of this can be found in Berends et al. (2018).

To compare the uncertainty of interventions we calculate the *relative uncertainty*, i.e. the uncertainty relative to the expected effect. For this, we use an adapted version of the coefficient of variation, defined as:

$$U_{r;90} = \frac{\text{MCI}_{90,\Delta H}}{\text{E}(\Delta H)} \qquad (5)$$

where $\text{MCI}_{90,\Delta H}$ is the 90% confidence interval for $\Delta H$ and $\text{E}(\Delta H)$ the average of $\Delta H$. The $U_{r;90}$ is expressed in percentages. Therefore, a $U_{r;90}$ of 100% means that the 90% MCI is as large as the average effect. Since both $\text{MCI}_{90,\Delta H}$ and $\text{E}(\Delta H)$ are stochastic, $U_{r;90}$ is stochastic as well.

## 3 Results

### 3.1 Pre-intervention uncertainty

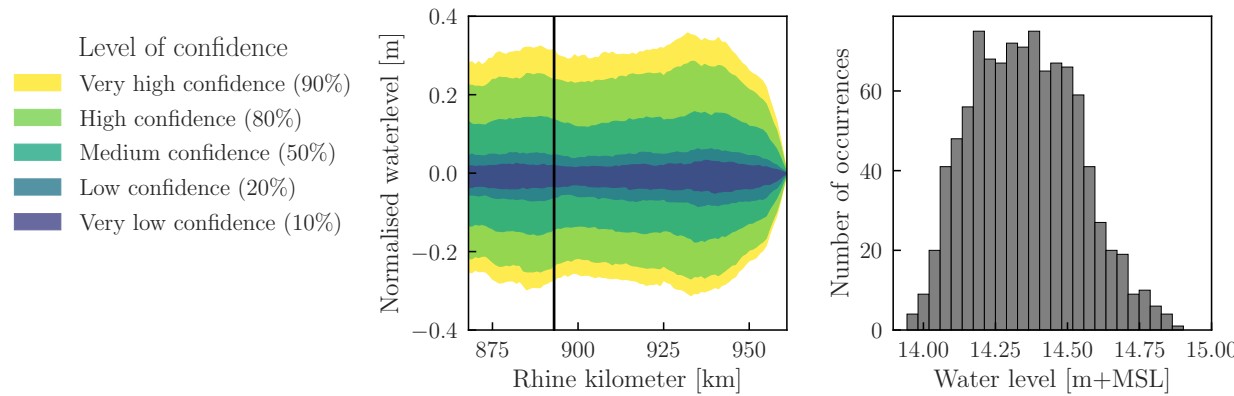

**Figure 5.** Middle panel: Confidence levels of design water levels, normalised by the average water levels at each location. The water flow is from left (upstream) to right (downstream). The black line indicates km 893. Right panel: the histogram of the (not normalised) water levels at km 893.

Figure 5 shows the uncertainty in the water levels along the River Waal before any intervention has taken place, normalised by the average water levels. The average confidence intervals over the entire stretch range from 55.4 cm for the 90% interval to 5.7 cm for the 10% interval. However, the intervals are not uniform along the river. At the downstream boundary the intervals collapse to zero due to the fixed boundary condition. Upstream from the boundary, there is a noticeably smaller interval, approximately between km 885 and km 925. This is attributed to the relatively wider floodplains in this part of the river, which reduces flow through the main channel (Warmink et al., 2013b). The right hand panel of Figure 5 shows the histogram at km 893, which enables visual comparison with Figure 6 in Warmink et al. (2013b), who performed their analysis with the



WAQUA modelling system for the 1995 River Waal and slightly higher upstream boundary discharge. They reported a 95% confidence interval of 68 cm (13.99 m + MSL to 14.66 m + MSL) at km 893. Our results also show a 95% confidence interval of 68 cm, although at slightly higher water levels (14.05 m + MSL to 14.73 m + NAP). This provides confidence in the model results. However, we highlight two differences with respect to the study of Warmink et al. (2013b). First, we did not include the

5 vegetation roughness model as a source of uncertainty, but we added vegetation parameter uncertainty. This change does not seem to have affected the output uncertainty of flood levels significantly. Second, the increase in flood levels with respect to the study of Warmink et al. (2013b) was not expected, given that between 1995 and 2015 a large scale flood mitigation programme (called 'Room for the River') was carried out aimed at reducing flood levels. However, given the different modelling systems and assumptions, this comparison cannot be used to judge the effectiveness of that programme.

### 3.2 Uncertainty of along-channel flood level decrease

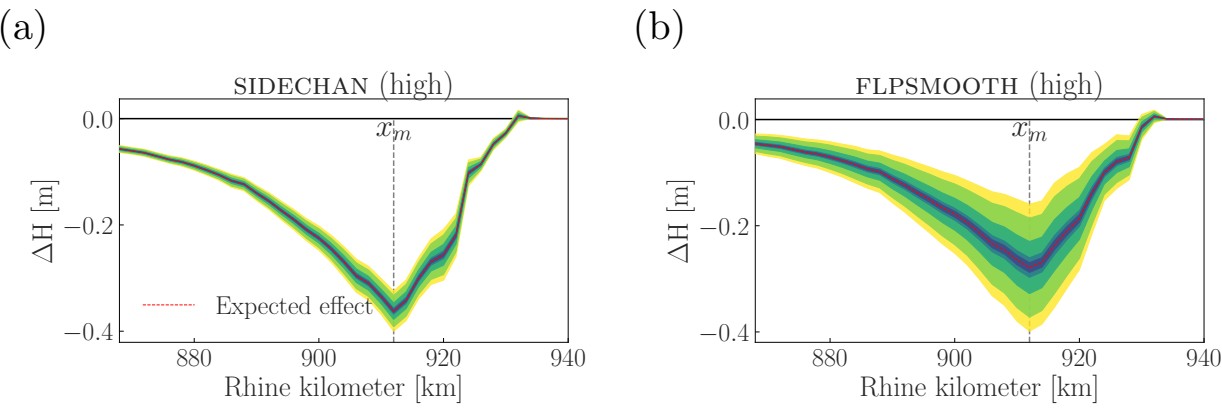

**Figure 6.** The expected lowering of flood levels for side channel construction (a, SIDECHAN), and floodplain smoothing (b, FLPSMOOTH), both high intensity and flow direction from left to right. The colours in this chart convey identical confidence levels as shown in Figure 5. The location of the maximum expected effect is indicated by the dashed line and annotation $x_m$.

The primary objective of the flood mitigation measures is to lower flood levels during a given design discharge. In deterministic approaches, this effect is typically communicated with an along-channel diagram showing the difference (i.e., before and after the intervention) in flood levels (see e.g. Figure 3 in Klijn et al., 2018). Figure 6 shows a similar diagram, augmented with information on the uncertainty of the prediction. The expected effect follows the general trend as a deterministic approach.

Starting from the downstream end of the intervention (around km 932), water levels gradually decrease due to the local increase of conveyance compared to the reference state. From km 912 upstream, water levels gradually return to the reference equilibrium. The gradual relaxation to equilibrium levels both over and upstream from the intervention, i.e. backwater curves,



is explained through basic sub-critical flow theory. We note that the length of the backwater curves spans tens of kilometres resulting in a residual water level decrease of approximately 5 cm near the upstream boundary (km 868). In theory, this could affect the distribution of discharge at the bifurcation of the River Waal with the Pannerden Canal. This secondary effect is not accounted for in this study.

The novelty of Fig. 6 these results is mainly found in the uncertainty, visualised through the confidence intervals. These intervals follow the movement of the average; going down when the average decreases and relaxing toward the reference equilibrium upstream from the location of maximum effect ($x_m$). This shows that taking into account parameter uncertainty does not fundamentally change established typical effects of flood mitigation measures. However, the movement with the average is not accompanied by parallelity: the range of the intervals is not constant along the river. Instead, the range of the

intervals scale with the expected effect: the ranges are small for small effects, and increase with increasing effect. Consequently, uncertainty is largest at the maximum effect. Downstream from the intervention the confidence bands tend to zero, since for a steady computation, there is no difference between the pre- and post-intervention models and no backwater effects. Upstream from the intervention the confidence intervals reduce to zero along with the backwater curve effect.

     Next, we compute the relative uncertainty ($U_{r;90}$) for each intervention and intensity. Since $U_{r;90}$ is stochastic, we first reduce

this by calculating the expected value ($E(U_{r;90})$) for each point along the river (Fig. 7). Results show that values for $E(U_{r;90})$ are fairly constant over the river length up to the location of the maximum flood level decrease ($x_m$) for all interventions. A notable exception is the peak at approximately km 885 for low-intensity GROYNLOW. This is attributed to small inaccuracies in the estimation of the expected effect. Since $U_{r;90}$ is a ratio, its value is sensitive to small values of the denominator, which is the expected effect (see Eq. (5)). GROYNLOW has a very minor effect (maximum of 1 cm at low intensity), so small inaccuracies

in estimation will greatly affect $U_{r;90}$. For this same reason, we do not compute $U_{r;90}$ upstream from the maximum effect, as the flood level decrease will rapidly return to and cross zero, which will lead to rapidly exploding $U_{r;90}$ values. From the fact that we do not observe such peaks much more often, we gain confidence that the chosen estimation method is suitable for this type of analysis even at very small-scale effects. In general, the constant values for $U_{r;90}$ shows that it is a useful parameter to characterise the uncertainty of an intervention, as the increase in uncertainty with the effect, observed in Figure 6 is sufficiently

summarised by a single $U_{r;90}$ value.

     It is interesting to note that in theory, the adaptation length (i.e. the length scale of the backwater curve) is affected by the equilibrium water level and can therefore result in converging or diverging confidence intervals. Since $E(U_{r;90})$ is relatively constant along the river, the adaptation lengths (i.e. the length scale of backwater effects) are not significantly affected by the intervention. Therefore, we see that all uncertainty is generated over the length of the intervention and subsequently diminishes

upstream as water levels return to their equilibrium. In other words, if the adaptation lengths are known, the uncertainty upstream of $x_m$ can be readily estimated from the relative uncertainty at $x_m$.

     A comparison of all interventions (Table 2) shows that small effects ($E(\Delta H)_{xm}$) are accompanied by small uncertainties ($E(MCI_{90})_{xm}$). This shows that even small changes to rivers can be predicted, even when the absolute uncertainty in water levels (see figure 5) is an order of magnitude larger than the expected effect.





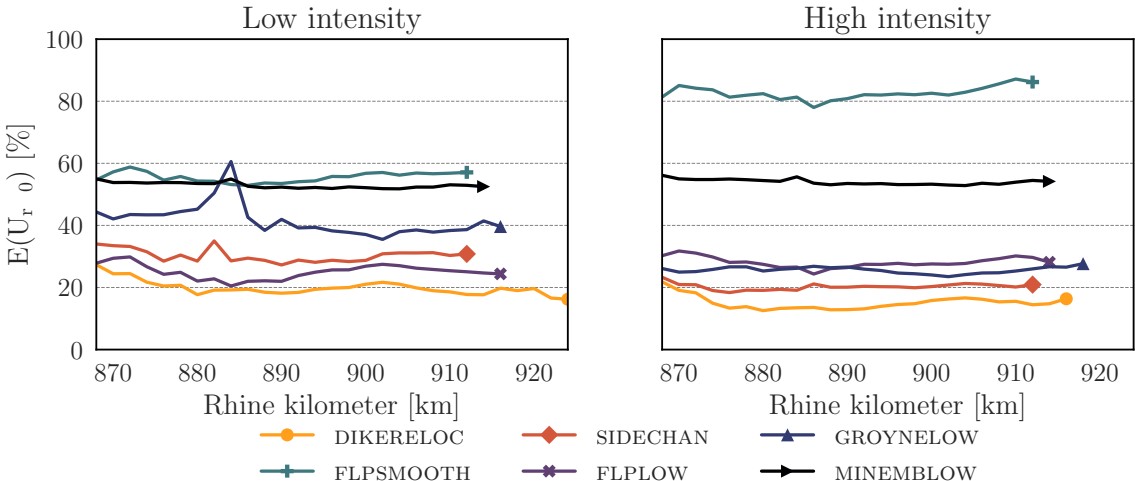

**Figure 7.** The expected values for the relative uncertainty ($U_{r;90}$) along the river, from the upstream boundary at km 868 to the location of maximum effect $x_m$ indicated by the marker, showing relatively constant values upstream from $x_m$.

**Table 2.** Summarised results for each measure. H=high intensity; L=low intensity; $x_m$= location in Rhine kilometre (km) where maximum effect occurs; $E(\Delta H)_{xm}$= expected effect at $x_m$; $E(\text{MCI}_{90})_{xm}$=expected 90% uncertainty at $x_m$; $\widehat{U_{r;90}}$= Coefficient of variation, averaged over the river length. The rows are sorted from highest expected effect (top) to lowest expected effect (bottom).

| Intervention | | $x_m$ | $E(\Delta H)_{xm}$ | $E(\text{MCI}_{90})_{xm}$ | $\widehat{U_{r;90}}$ |
|---|---|---|---|---|---|
| DIKERELOC | H | 916 | -1.08 | 0.176 | 15% |
| | L | 924 | -0.21 | 0.034 | 20% |
| FLPLOW | H | 914 | -0.80 | 0.224 | 28% |
| | L | 916 | -0.10 | 0.024 | 25% |
| SIDECHAN | H | 912 | -0.36 | 0.076 | 18% |
| | L | 912 | -0.02 | 0.007 | 20% |
| FLPSMOOTH | H | 912 | -0.28 | 0.239 | 82% |
| | L | 912 | -0.04 | 0.023 | 55% |
| GROYNLOW | H | 918 | -0.04 | 0.012 | 23% |
| | L | 916 | -0.01 | 0.004 | 35% |
| MINEMBLOW | H | 914 | -0.03 | 0.014 | 53% |
| | L | 914 | -0.03 | 0.013 | 52% |

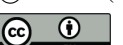



### 3.3  Inter-comparison between interventions

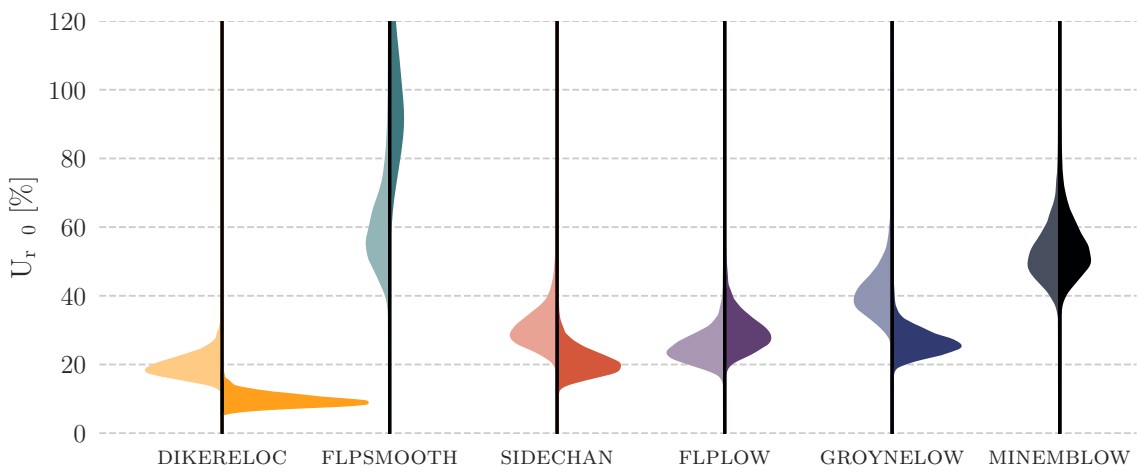

**Figure 8.** Probability densities of the relative uncertainty ($U_{r;90}$) at the location of maximum effect ($x_m$), for all interventions, showing that FLPSMOOTH is significantly more uncertain than other interventions. Most interventions show similar distributions for the low intensity (left, light coloured) and high intensity (right, dark coloured) variants.

A straightforward way to compare differences between the effect of two or more interventions is to look at the backwater figures. In Fig. 6 we see two diagrams of interventions that result in flood level decrease within the same order of magnitude, but with very different uncertainty. The uncertainty of FLPSMOOTH is much higher than that of SIDECHAN, as represented by
the wider confidence intervals.

To systematically compare interventions we compute the relative uncertainty $U_{r;90}$ at the location of the maximum effect for each intervention (Figure 8). Since $U_{r;90}$ is stochastic, we visualised the kernel density estimation of their probability distributions. In general we observe that (i) FLPSMOOTH and MINEMBLOW are clearly more uncertain than the other four interventions and (ii) that the low-intensity variants have similar distributions compared to their high-intensity variants, both in
shape and location. Even in the cases where the high-intensity variant distribution is clearly different from the low-intensity variant (DIKERELOC, GROYNLOW and FLPSMOOTH), the distribution still overlap. This shows that the intensity of an intervention is not a deciding explaining factor for $U_{r;90}$, even though it is evidently important in explaining the absolute uncertainty (MCI$_{90}$) and the expected effect. The latter follows from the earlier observation that the absolute uncertainty scales with the expected effect following a (constant) relative uncertainty.

For decision making, it may be useful to estimate how much uncertainty is expected for a given expected effect. When considering different flood mitigation measures, the choice may depend on the level of acceptable uncertainty. By linearly interpolating between the two low and high intensities, we can estimate the uncertainty for a given expected effect for each



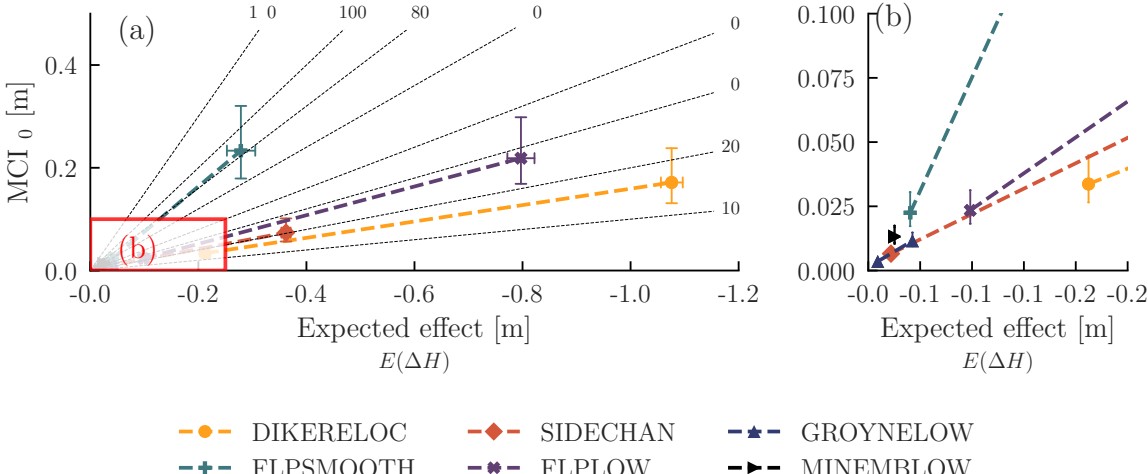

**Figure 9.** The expected effect ($E(\Delta H)$) against the 90% confidence interval, linearly interpolated between the low and high intensities. The relationship between the two is described by the relative uncertainty $U_{r;90}$, here shown as dashed grey lines. The right hand panel (b) displays a zoomed-in part of the left-hand graph.

intervention (Figure 9). For example, consider that the objective of a planned flood mitigation measure is 25 cm. We see from figure 9 that this can be accomplished by four different interventions (FLPSMOOTH, FLPLOW, SIDECHAN and DIKERELOC), within the bounds marked by the low- and high intensities. The confidence intervals differ markedly between those choices. Under the assumption of linear interpolation between the two intensities, the 90% confidence intervals for an expected decrease

of 25 cm are estimated at 4 cm (DIKERELOC), 5 cm (SIDECHAN), 7 cm (FLPLOW) and 21 cm (FLPSMOOTH). These intervals may or may not be acceptable.

Instead of looking at the expected effect, the confidence intervals can be marginalised by designing for a given exceedance probability of the effect (Figure 10). The desired effect is related to the expected effect through the exceedance probability. For example, figure 10 (a) shows various exceedance probabilities for FLPSMOOTH, linearly interpolated between the low-intensity

and high-intensity variants as function of the expected effect. The high-intensity variant of FLPSMOOTH has an expected flood level decrease of 28 cm. However, the likelihood that the decrease is smaller (or higher) than that number is only approximately 50%. This means that the expected effect of 28 cm is as likely as not to be met by the proposed intervention. In contrast, the 90% exceedance probability for the high-intensity variant is 18.7 cm. This means that it is "very likely" (following IPCC terminology, see Solomon et al. (2007)) that the flood level decrease is larger than 18.7 cm.

This approach can be applied to the earlier example of a planned mitigation measure with a 25 cm flood level decrease objective. In figure 10 (b), we see that at an exceedance probability of 66% ("Likely"), all four possible interventions can be implemented. However, to meet this likelihood all interventions have to be "over-designed", i.e. to meet a larger expected flood level decrease than the given objective. For example, floodplain smoothing needs to be designed for an expected effect





of 28 cm for a 66% likelihood of reaching the objective of 25 cm. The amount by which the measure needs to be over-designed depends on the uncertainty of that measure and the exceedance probability. Higher exceedance probabilities such as a 95% likelihood ("Extremely likely") can also preclude some flood mitigation measures (Figure 10 (c)). With this level of confidence, FLPSMOOTH can no longer be considered: none of the computed examples reach an 'extremely likely' flood level

decrease of 25 cm within the limits of the considered interventions. In this case, there is a physical limit to an even higher intensity FLPSMOOTH, since there is simply no more available land within the study area to turn into meadow. Extrapolation beyond the given limits is therefore not possible, although a higher effect could be obtained by simply enlarging the study area. Within the given limits, the three remaining interventions (DIKERELOC, SIDECHAN and FLPLOW) can still be used. They would need to be designed for an expected effect of -0.27m , -0.28m and -0.29m, respectively. For perspective, it is worthwhile

to point out that one of the greatest interventions on the River Waal in the past decades, the Nijmegen-Lent dike relocation project, has (deterministically) predicted effect of -27 cm, at an estimated cost of more than ten million Euro per cm flood level decrease.

## 4    Discussion

The results show that some interventions are inherently more uncertain than other interventions. Given the complexity of high-

detailed modelling of floodplains, it is not straightforward to explain these differences. The statistical causes of uncertainty are discussed in detail by Berends et al. (2018). They observed that the removal of existing stochastic elements (e.g., a vegetation polygon) in exchange for new ones greatly the unexplained variance ($\epsilon$ in (eq. 3)) and therefore increases effect uncertainty. We see that here as well: FLPSMOOTH, which removes high-friction vegetation in exchange for meadow, is by far the most uncertain intervention. However, in this study we have argued that the relative uncertainty ($U_{r;90}$) is a more useful parameter

than the absolute uncertainty (MCI). In that context, a large absolute uncertainty can be offset by a large expected flood level decrease. This is evidenced by both DIKERELOC and FLPLOW, which both have a large absolute uncertainty at high intensity but a relatively low relative uncertainty. A reduction in relative uncertainty could therefore be achieved by minimising the amount of change to existing floodplains while optimising the expected flood level decrease. This general observation is likely applicable to other rivers and case studies, although the specific results for these interventions is likely not.

In studying the effect of uncertainty on intervention design (figures 9 and 10) we linearly interpolated between the high intensity and low intensity interventions. The assumption of linearity is reasonable given the approximately constant $U_{r;90}$ between the two intensities. However, in practice this should be only seen as a first approximation. Multiple design iteration would be necessary to arrive at a design that meets the requirements.

A lack of computational resources is often named as the main obstacle that motivates the use of deterministic approaches

over probabilistic approaches. The analysis performed in this study is nonetheless feasible due to the CORAL approach, which by reducing the number of model evaluations, reduces the required computer resources. An additional advantage of this method is that the uncertainty of the estimation is known and can be explicitly incorporated in analyses. It is possible to decrease this estimation uncertainty by increasing the number of model runs with the post-intervention models (Berends et al., 2018), if


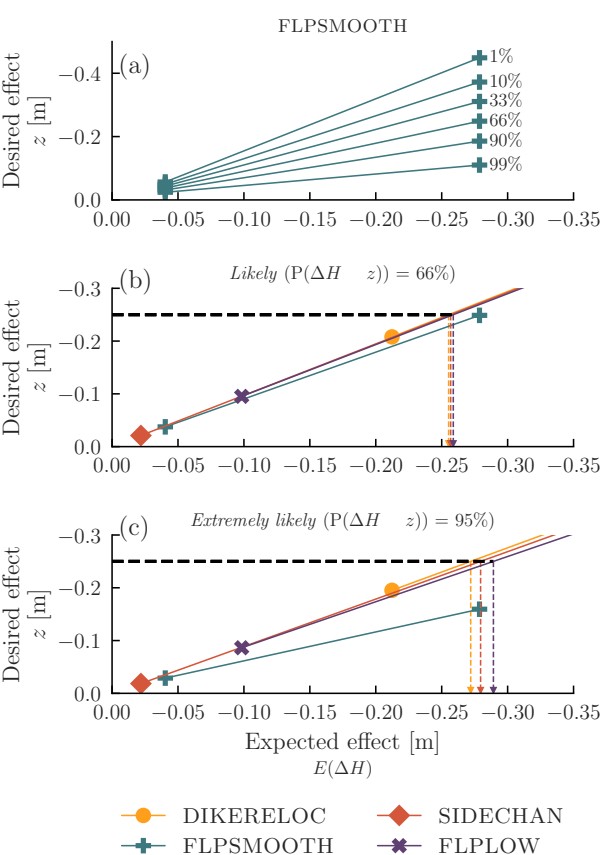

**Figure 10.** (a) The relationship between the expected effect $E(\Delta H)$ and the exceedance level $z$ for various exceedance probabilities. (b) Inter-comparison of a 66% exceedance probability for four interventions. The dashed black line indicates a $z$ of 25 cm, and the coloured dashed lines indicate which expected level belong to that $z$, given the exceedance probability (see text). (c) same as (b), but for an exceedance probability of 99%.



(computational) resources allow. In this study, we show that the relative uncertainty $U_{r;90}$ can be estimated to a sufficient precision (Fig. 8) to draw conclusions, even with a relatively low number of model runs (20 for each intervention).

Our approach is a form of forward uncertainty analysis: the quantification of model output uncertainty based on uncertainty in model assumptions without taking into account historical evidence of the goal variable (i.e., water level measurements).

We justify choosing this approach based on the assumption that no historical information can be used to infer or narrow the distributions of the uncertain parameters, because (a) such information does not exist for our study case and (b) measurements after the intervention will never be available *ex ante*. The results of our analysis are therefore conditional on our assumptions about the validity of the model, the selection uncertainty sources and the probability distributions of these sources. We have taken care to select the stochastic variables, and their distributions, based on previous research specific to our study case.

Forward uncertainty analysis is contrasted (e.g., see Beven et al., 2018) with so-called inverse methods, such as GLUE (Beven and Binley, 2014), DREAM (Vrugt et al., 2008) or deterministic calibration, that use historical evidence to potentially narrow the uncertainty bands of input variables. However, the potential use of historical data to constrict uncertainty faces complex challenges which will have to be met. Chief among them are extrapolation to unmeasured conditions and dealing with the large number of uncertain variables (here, 39) which may not be uniquely identifiable (Werner et al., 2005). In any case,

like traditional deterministic approaches, the assumptions underlying probabilistic approaches should be readily available, transparent, and open to discussion.

In literature, there are several examples of probabilistic and risk-based approaches in designing flood adaptation strategies (Lendering et al., 2018; Klijn et al., 2018). Model uncertainty is generally not explicitly quantified as part of such approaches, but could be a natural addition to such approaches. This is especially so when model uncertainty influences intervention design

(Hall and Solomatine, 2008; Beven et al., 2018). The quantification and visualisation of uncertainties here may be a step towards this goal.

## 5 Conclusions

In general, our study shows that explicitly quantifying the uncertainty of predicted flood mitigation measures provides valuable information to decision makers and modellers. On the one hand, results show how taking into account uncertainty can lead

to different design choices. On the other hand, even small effects on flood levels can be quantified, because small effects are accompanied by small uncertainties. This shows that model uncertainty does not invalidate model-supported decision making in river management, but enriches it.

Our first objective was to quantify the effect of parameter uncertainty on the predicted effectiveness of flood mitigation measures. Based on previous studies, we quantified the parameter uncertainty for 39 variables and estimated the uncertainty of

model output. Results show that the absolute uncertainty of the predicted effect of flood level decrease is highly dependent on the type of intervention and location along the river. However, we found that the confidence bounds of flood level decrease along the river can be adequately described by a single "relative uncertainty" metric, defined as the ratio between the 90% confidence interval and the expected effect. This ratio remains relatively constant along the river and between intensities of intervention



types, and enables us to make some general observations. First, all uncertainty is 'generated' where the intervention has modified the river system. Upstream from there, the uncertainty gradually diminishes upstream with a constant rate following typical backwater curves. Second, a higher expected flood level decrease lead to a higher uncertainty, and a small flood level decrease was accompanied by a small uncertainty. The ranges of the expected relative uncertainty varied between 15% and

40% for most interventions, which means that the size of the 90% confidence bounds of those interventions is less than half of the expected flood level decrease.

The second objective was to establish a relationship between the expected effect and its uncertainty, to aid in the decision making process. We observe that interventions of different types may reach the same expected flood level decrease, but having different uncertainty. Specifically, a large-scale but relatively ineffective intervention such as floodplain smoothing (by

removing high-friction vegetation) has a high relative uncertainty compared to alternative interventions. The intensity of an intervention (e.g. total area of vegetation smoothed) may be increased to reach a higher effect. Our results show that higher intensity also leads to a higher uncertainty, while the relative uncertainty remains approximately constant. This relationship was then used to show how explicit uncertainty quantification and differences in relative uncertainty between various interventions may affect design choices, depending on the level of acceptable uncertainty. For a fixed level of acceptable uncertainty (i.e.,

by a given exceedance probability), we graphically demonstrated that interventions need to be designed for a larger expected flood level decrease than the given objective.

*Code availability.*  In this article we used the following code and software. For the hydraulic modelling we used Delft3D Flexible Mesh (FM) version 1.1.261.52670. Delft3D FM is available from (https://oss.deltares.nl/web/delft3dfm). For the generation of the various flood mitigation measures we used the Python tool RiverScape (for availability see Straatsma and Kleinhans (2018)). The uncertainty quantification

method CORAL is available from GitHub (https://github.com/kdberends/coral).

**Appendix A: Parameter distributions for vegetation types**

*Competing interests.*  The authors declare that they have no conflict of interest.

*Acknowledgements.*  This study is part of the research programme RiverCare, supported by the domain Applied and Engineering sciences (AES), which is part of the Netherlands Organisation for Scientific research (NWO), and which is partly funded by the Ministry of Economic

Affairs, under grant number P12-14 (Perspective programme). This study benefitted from cooperation within the Netherlands Centre for River studies (NCR).



**Table A1.** Parameters of the floodplain roughness class distributions. Codes marked with an asterisk * are not used in the Waal model

| Class Code | Name | Parameters | | | |
|---|---|---|---|---|---|
| | **Empirical distribution** | | | | |
| 612 - 637 | Alluvial bed | | | | |
| | **Uniform distribution** | | | | |
| n/a | Classification map | | | | |
| | **Triangular distributions** | *min* | *mean* | *max* | |
| 102 | Deep bed | 0.025 | 0.03 | 0.033 | |
| 104 | Natural side channel | 0.03 | 0.035 | 0.04 | |
| 105 | Side channel | 0.025 | 0.03 | 0.033 | |
| 106 | Pond / Harbor | 0.025 | 0.03 | 0.033 | |
| 111 | Sand bank | 0.025 | 0.03 | 0.033 | |
| 121 | Field | 0.02 | 0.03 | 0.04 | |
| | **Lognormal distributions** | $\mu_{h_v}$ | $\sigma_{h_v}$ | $\mu_{n_v}$ | $\sigma_{n_v}$ |
| 1201 | Production meadow | -3.18 | 0.47 | 2.40 | 0.77 |
| 1202 | Natural grass and hayland | -0.74 | 0.53 | -2.64 | 0.93 |
| 1203 | Herbaceous meadow | -1.64 | 0.32 | 2.59 | 0.33 |
| 1211* | Thistle herb. Veg. | -1.29 | 0.33 | 1.05 | 0.43 |
| 1212 | Dry herbaceous vegetation | -0.59 | 0.39 | -3.06 | 0.65 |
| 1213* | Brambles | -0.67 | 0.21 | -0.73 | 0.36 |
| 1214* | Hairy willowherb | -1.89 | 0.56 | -0.25 | 0.49 |
| 1215* | Reed herb. Veg. | 0.60 | 0.22 | -1.83 | 0.27 |
| 1221 | Wet herb. Veg. | -1.08 | 0.38 | -1.49 | 0.44 |
| 1222* | Sedge | -1.32 | 0.67 | 0.04 | 0.63 |
| 1223 | Reed-grass | -0.92 | 0.86 | -2.19 | 0.16 |
| 1224 | Bulrush | -0.81 | 0.67 | 0.04 | 0.63 |
| 1225* | Reed-mace | 0.37 | 0.23 | -1.12 | 0.57 |
| 1226 | Reed | 0.94 | 0.13 | -1.14 | 0.42 |
| 1231 | Softwood shrubs | 1.81 | 0.24 | -2.20 | 0.79 |
| 1232 | Willow plantation | 1.05 | 0.43 | -3.23 | 0.62 |
| 1233 | Thorny shrubs | 1.48 | 0.64 | -1.73 | 0.41 |
| 1241* | Hardwood production forest | *Deterministic* | *Deterministic* | -4.68 | 0.67 |
| 1242 | Softwood production forest | *Deterministic* | *Deterministic* | -4.72 | 0.66 |
| 1243* | Pine forest | *Deterministic* | *Deterministic* | -4.18 | 0.54 |
| 1244 | Hardwood forest | *Deterministic* | *Deterministic* | -3.45 | 0.77 |
| 1245 | Softwood forest | *Deterministic* | *Deterministic* | -3.04 | 0.99 |
| 1246 | Orchard low | 1.10 | 0.10 | -3.72 | 0.25 |
| 1247 | Orchard high | 1.78 | 0.21 | -4.61 | 0.12 |
| 1250 | Pioneer vegetation | -2.87 | 0.18 | -1.93 | 0.50 |



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
