# Peer review of "Uncertainty quantification of flood mitigation predictions and implications for interventions"

_Natural Hazards and Earth System Sciences, 2018_

## Referee Comment (RC1) · Guillaume (Referee) · 12 Dec 2018

This article assesses the impact of 12 interventions on flood water levels in the Waal River, including estimation of uncertainty. It goes further than most other studies in 1) estimating uncertainty in uncertainty attributable to the methods used, 2) discussing the potential to approximate the uncertainty, in ways that make the results of the analysis more accessible to a broader audience. The article is well written and clear. The issues I would like to highlight are relatively minor - clarifying some claims rather than fundamentally questioning them. I do suggest some small extensions to the analyses to help in clarifying these claims.

[Figure]

Specific comments —————

I would rather that the abstract avoid emphasising that "relative uncertainty" is a "newly introduced" metric. As the authors do acknowledge in the main text, it is closely related to the coefficient of variation. If this metric is kept as a contribution, it would require greater discussion of why is preferred over the CV. The greater contribution here, in any case, is that relative uncertainty is identified as a key parameter to obtain approximations of uncertainty in this type of problem.

p9 L25 "We would need to obtain both the model output probability distributions, as well as the covariances between the two distributions" This is not entirely true - most of the parameters would be identical in both states of the river, so the difference can be calculated directly as a single distribution - covariances probably do not need to be calculated. The authors do end up doing this with the "union stochastic parameter space" (p11), so it seems worthwhile to clarify that you do not explicitly estimate covariances. It is still correct that this is probably not tractable for 12 interventions.

p15 To support the claim that values are relatively constant, it would be useful to show the range of E(Ur90) over the length of the river, either in Figure 7 or Table 2 E(range(Ur90)) would also be useful, given it is available, i.e. the range of Ur90 over the length of the river, averaged over all realisations There is enough variation there assuming relative uncertainty is constant seems like it would still be a substantial simplification.

p16 Figure 8 "FLPSMOOTH is significantly more uncertain" The use of the word "significantly" suggests this is a statistical statement. It might be worth putting a p-value on this, given you have uncertainty on the uncertainty. Visually, it seems high intensity FLPSMOOTH could be significantly more uncertain (p=0.05), but that low intensity is comparable to groyneflow or minemblow, also confirmed by Figure 7. L6-14 could also comment on statistical significance of differences in uncertainty.

p17 Figure 9 This figure is illustrative of the interpolation method, but currently only

hints at how good the interpolation method would be. You have two sources of information on this: 1) the uncertainty in MCI90 and E(dH) 2) the variation in these values along the river The error bars for (1) in the figure do suggest that there is quite a bit of uncertainty involved in the interpolation. I would suggest showing a ribbon around the line encompassing all potential lines that could be interpolated using 1 and 2. This doesn't detract from the utility of using linear interpolation, but would provide a better sense of the confidence one should have in using this simplification. If the uncertainty due to using CORAL is too large, perhaps more runs might also help make your point more effectively.

p17- exceedance probabilities If I understand correctly, this now linearly interpolates exceedance probabilities rather than relative uncertainty. It's not clear why this is a permissible inference. Are there characteristics of the distributions that allow this jump? What makes (all!) exceedance probabilities linear if relative uncertainty is constant with regard to expected effect? I assume at least some kind of symmetry comes into play? Even if the assumption of linearity doesn't end up holding completely, I would be in support of keeping this analysis in the paper, with some representation of the uncertainty in uncertainty involved (as for Figure 9). I think it's very important to be able to say something about the extremes of the distribution, not just about the size of the uncertainty.

p19 Discussion It would be useful to mention that the study only considered individual interventions, whereas it would be possible to combine measures in practice. I assume the combined effect is unlikely to be a linear combination of individual effects, so I don't think this study can be used to support any claims regarding such combinations. It would, however, be useful to highlight it as future work and speculate about possible issues that might crop up.

Technical comments ——————-

Table 1 The link with subsections in 2.3 could perhaps be emphasised, e.g. by using

consistent ordering and grouping terms

p7 L14 "general extreme values distribution" Should be "Generalized Extreme Value distribution"?

p7 L20 "a 95% confidence limits at 0.31 m and 1.0 m." drop the "a"

p14 Table 2 Should "coefficient of variation" be average relative uncertainty E(Ur90)? Otherwise, this raises the question of whether relative uncertainty was indeed needed?

Figure 7 y axis label should read Ur90? (9 is missing)

Figure 9 also seems to have some text rendering issues

p18 L17 "greatly the unexplained variance" Word missing (increased?)

---

## Referee Comment (RC2) · Kwakkel (Referee) · 19 Dec 2018

Overall the paper is well structured, well written and quite clear. However, I have a number of reservations regarding the manuscript. Several of these overlap with the points raised by Joseph in his comments, so I won't discuss those here in any further detail.

My main reservation is regarding the contribution that the paper is trying to make. In the abstract the authors in the second sentence mention the role of model-based analyses in supporting decision making, while ending the abstract with claims about model-based decision making. In the middle of the abstract, the authors mention a

new metric 'relative uncertainty'. Next, they argue that using this new metric they can provide insight into the uncertainty about the effects of a variety of flood risk reduction intervention.

I have two reservations here. First, regarding the claims related to decision making. In my view, these claims are not well developed in the paper. Outside of the introduction and conclusion, the term itself only appears once. Moreover, if I look at the presented results and their visualization, I doubt these would be used by a decision maker or even someone directly advising a decision maker. Rather, the presented analyses are useful for people working on the design of flood risks reduction strategies, while some of the results might indirectly be used in decision making. As such, I would suggest removing the term from the title as well as tone done any other claims made in the abstract and introduction. Alternatively, the authors would need to expand their discussion in the main text on how providing insights into the range of uncertainty about expected effects of measures can assist decision making on flood risk. This, however, requires discussing notions such as robustness (see e.g., McPhail et al., 2018) and flexibility, as well as discussions on well characterized uncertainty (i.e., you have a meaningful pdf) and Knightian or deep uncertainty (i.e., for whatever reason you don't have a meaningful and uncontested pdf).

My second reservation is with the new metric itself. As also indicated by Joseph, this metric is closely related to the coefficient of variation. Moreover, the more theoretical discussion of this new metric is confined to one short paragraph around equation 5. Too play devils advocate: what is the merit of publishing a paper whose only contribution is a single equation closely related too an already established metric? The case study mainly serves to establish the value of this metric, while the models are taken from earlier work. If you insist on having the metric as a key contribution, than a comparative perspective would be more appropriate. So, what other metrics already exist that could serve a similar function? Classic robustness metrics as well as the coefficient of variation would be logical candidates. How is this metric different from these,

and what are the relative merits of the new metric relative to the others?

Basically, both reservations have to do with how the authors currently position their work. I am having profound reservations regarding the first two paragraphs on page 3. First, the authors cite Pinter on fuzzy math, suggesting that not providing explicit uncertainty quantification leaves decision makers free to interpret the uncertainty in any way they like. My argument would be that the converse also often happens. As for example elaborated nicely in Pilkey and Pilkey-Jarvis (2007), a lot of quantitative analysis for supporting decision making becomes useless arithmetic that is used strategically. In a different line of literature, researchers working on post normal science often claim that many models have not even one significant digit. That is, models give only a false sense of precisions. A third line of research, as exemplified by Sarewitz (2004) shows how more research and increasing efforts to quantify uncertainty can often make environmental controversies worse. In short, the relationship between models, model results, uncertainty about models results and decision making processes is quite a bit more complicated than the authors seem to think. It would benefit the manuscript it the claimed contribution is better positioned relative also to these strands of literature.

Minor remarks

Might it not be more convenient to show the interventions (2.2.1-2.2.6) in a table?

What is the runtime of the detailed model for a single run?

Page 18, line 17, "for new ones greatly the unexplained" I guess some words are missing here

Page 17, line 4, partly overlapping with Joseph's comment, but what justifies the linear interpolation?

McPhail, C., Maier, H. R., Kwakkel, J. H., Giuliani, E., Castelletti, A., & Westra, S. (2018). Robustness metrics: How are they calculated, when should they be used and why do they give different results? Earth's Future. doi:10.1002/2017EF000649 Pilkey,

O. H., & Pilkey-Jarvis, L. (2007). Useless Arithmetic: Why Environmental Scientists Can't Predict the Future. New York, USA: Columbia University Press. Sarewitz, D. (2004). How science makes environmental controvesies worse. Environmental Science & Policy, 7, 385-403. doi:10.1016/j.envsci.2004.06.001

---

## Author Response (AR1)

**Combined point-by-point reply to the comments**

*To reviewer comments for discussion paper https://doi.org/10.5194/nhess-2018-325*
*Berends, K.D., Straatsma, M.W., Warmink, J.J., Hulscher, S.J.M.H., 17 April, 2019*

All referee comments are numbered by referee number and comment number, e.g. [R1.2] for the second comment of the first referee. The author's response is colour coded:

Black - Reviewer comment
Blue - Author reply
Green - Proposed change in manuscript

**Replies to referee 1**

We would like to thank the reviewer for taking the time to review our manuscript. We highly appreciate his suggestions and comments, which are helpful in improving the manuscript. Below we have replied to the various comments made by the reviewer.

[R1.1] This article assesses the impact of 12 interventions on flood water levels in the Waal River, including estimation of uncertainty. It goes further than most other studies in 1) estimating uncertainty in uncertainty attributable to the methods used, 2) discussing the potential to approximate the uncertainty, in ways that make the results of the analysis more accessible to a broader audience. The article is well written and clear. The issues I would like to highlight are relatively minor - clarifying some claims rather than fundamentally questioning them. I do suggest some small extensions to the analyses to help in clarifying these claims.

[Reply to R1.1] We are grateful for the kind words, and the acknowledgement of the reviewer that our efforts to quantify uncertainty by approximation makes uncertainty quantification more accessible to a broader audience.

[R1.2] I would rather that the abstract avoid emphasising that "relative uncertainty" is a "newly introduced" metric. As the authors do acknowledge in the main text, it is closely related to the coefficient of variation. If this metric is kept as a contribution, it would require
greater discussion of why is preferred over the CV. The greater contribution here, in any case, is that relative uncertainty is identified as a key parameter to obtain approximations of uncertainty in this type of problem.

[Reply to R1.2] We agree with the reviewer that the introduction of relative uncertainty as a metric is not the key contribution of this paper, rather that the relative uncertainty is an effective metric to discuss the uncertainty of engineering interventions, and that this should be reflected in the abstract. In the abstract, we wrote:

" (...) We show that the uncertainty of an intervention can be adequately described by the newly introduced `relative uncertainty' metric, defined as the ratio between the confidence interval and the expected effect. (...)"

We changed this sentence to:

[P1 L8-9 in modified manuscript]
"(...) We identified the relative uncertainty, defined as the ratio between the confidence interval and the expected effect, as a useful metric to compare uncertainty between different interventions. (...)"

[R1.3] p9 L25 "We would need to obtain both the model output probability distributions, as well as the covariances between the two distributions" This is not entirely true - most of the parameters would be identical in both states of the river, so the difference can be calculated directly as a single distribution - covariances probably do not need to be calculated. The authors do end up doing this with the "union stochastic parameter space" (p11), so it seems worthwhile to clarify that you do not explicitly estimate covariances. It is still correct that this is probably not tractable for 12 interventions.

[Reply to R1.3] The reviewer is correct in noting the covariances are not explicitly computed (or estimated). To be clear, we are not referring to the covariances between the stochastic parameters (which are in our case identical between the two states of the river), but the covariances of the distributions of the computed water levels. These should always be considered, either explicitly (if done analytically) or implicitly. Otherwise, the uncertainty of the intervention will be needlessly overestimated. Furthermore, the problem is *analytically* intractable because of the numerical models involved, regardless of any covariance.

We wrote [p9, L23-L27]:

"To compute differences between these two (stochastic) states of the river system, we would need to obtain both the model output probability distributions, as well as the covariances between the two distributions (Berends et al., 2018). This problem is not analytically tractable, but can be solved numerically by Monte Carlo simulation (MCS; Metropolis, 1984; Stefanou, 2009)."

We changed these sentences to:

[p9, L18-21 in modified manuscript]

"To analytically compute differences between these two (stochastic) states of the river system, we would need to obtain both the model output probability distributions, as well as the covariances between these two distributions. This problem is analytically intractable, but can be solved numerically by Monte Carlo simulation (MCS; Metropolis, 1984; Stefanou, 2009), while accounting for covariance by sampling from a union stochastic parameter space

(Berends et al., 2018). In the numerical approach, covariances are not explicitly computed."

[R 1.4] p15 To support the claim that values are relatively constant, it would be useful to show the range of E(Ur90) over the length of the river, either in Figure 7 or Table 2 E(range(Ur90)) would also be useful, given it is available, i.e. the range of Ur90 over the length of the river, averaged over all realisations There is enough variation there assuming relative uncertainty is constant seems like it would still be a substantial simplification.

[Reply to R 1.4] The first suggestion of the reviewer is to add the range of $E(U_r; 90)$ to either Fig. 7 or Table 2. In response, we will add the standard deviation of E(Ur90) to Table 2 to explicitly show the variation along the channel. In the text, we wrote

[p14, L14-L16]:
"Next, we compute the relative uncertainty $(U_r; 90)$ for each intervention and intensity. Since $U_r; 90$ is stochastic, we first reduce this by calculating the expected value $(E(U_r; 90))$ for each point along the river (Fig. 7). Results show that values for $E(U_r; 90)$ are fairly constant over the river length up to the location of the maximum flood level decrease (xm) for all interventions."

We changed these sentences to refer to the new information in Table 2:

[p14, L8-11]
"Next, we compute the relative uncertainty $(U_r; 90)$ for each intervention and intensity. We first marginalise the uncertainty in $U_r; 90$ (which is due to CORAL estimation) by calculating the expected value $(E(U r; 90))$ for each point along the river (Fig. 7). Figure 7, as well as the relatively small standard deviations in Table 2, show that values for $E(U_r; 90)$ are fairly constant over the river length up to the location of the maximum flood level decrease (x m ) for all interventions."

The second suggestion of the reviewer, is to show $E(range(U_r; 90))$, as well. This would show the variation in the estimation uncertainty along the channel. In figure 8, we visualized the estimation uncertainty at $x_m$. We feel that including variation along the channel as well, would conflate the estimation uncertainty with the variation along the channel and as such not provide meaningful additional information, while making the figure more difficult to understand.

[R 1.5] p16 Figure 8 "FLPSMOOTH is significantly more uncertain" The use of the word "significantly"
suggests this is a statistical statement. It might be worth putting a p-value
on this, given you have uncertainty on the uncertainty. Visually, it seems high intensity
FLPSMOOTH could be significantly more uncertain (p=0.05), but that low intensity is

comparable to groyneflow or minemblow, also confirmed by Figure 7. L6-14 could also comment on statistical significance of differences in uncertainty.

[Reply to R1.5] Here, it was not our intention to provide a statistical significance of differences between the distributions, given that that the answer from any statistical test used to obtain a p-value would be conditional on both the intervention subsample size and the number of MCMC steps used in the approximation. To prevent confusion, we removed the word 'significant'.

We wrote:

[caption Figure 8, p 16]:

"(...) showing that /flpsmooth/ is significantly more uncertain than other interventions."

We changed this to:

[caption Figure 8, p 16 in modified manuscript]

"(...) showing that /flpsmooth/ is more uncertain than other interventions."

[R1.6] p17 Figure 9 This figure is illustrative of the interpolation method, but currently only hints at how good the interpolation method would be. You have two sources of information on this: 1) the uncertainty in MCI90 and E(dH) 2) the variation in these values along the river The error bars for (1) in the figure do suggest that there is quite a bit of uncertainty involved in the interpolation. I would suggest showing a ribbon around the line encompassing all potential lines that could be interpolated using 1 and 2. This doesn't detract from the utility of using linear interpolation, but would provide a better sense of the confidence one should have in using this simplification. If the uncertainty due to using CORAL is too large, perhaps more runs might also help make your point more effectively.

[Reply to R1.6] We agree with the reviewer that there is quite a bit of uncertainty involved in the interpolation and that multiple lines could be drawn between the high and low intensities. In fact, showing a ribbon around the line would still not resolve all uncertainty, as this still is conditional on the assumption of a linear model (i.e. a ribbon would show parameter uncertainty of the interpolation model, but not the model uncertainty of the interpolation model structure).

The main purpose of this figure, as with Figure 10 (see reviewer comment R1.7), is to show how uncertainty implicates intervention design. We believe that introducing more complexity, by showing every possible line that could be drawn (under the assumption of the linear model), would detract from this message. On the other hand, it should be clearly stated that the linear line is, at best, a first approximation. We made two changes in the manuscript to reflect this.

In the results section we wrote

[p16, L16 – p17, L1]:

"By linearly interpolating between the two low and high intensities, we can estimate the uncertainty for a given expected effect for each intervention (Figure 9)."

We changed this sentence to:

[p16, L11 – p12, L1 in modified manuscript]

"By linearly interpolating between the expected values of the low and high intensities, we obtain a first approximation of the uncertainty for a given expected effect for each intervention (Figure 9)."

In the discussion we wrote

[P18, L25-28]:

"In studying the effect of uncertainty on intervention design (figures 9 and 10) we linearly interpolated between the high intensity and low intensity interventions. The assumption of linearity is reasonable given the approximately constant $U_{r;90}$ between the two intensities. However, in practice this should be only seen as a first approximation. Multiple design iteration would be necessary to arrive at a design that meets the requirements."

We changed and expanded upon this section:

[p20, L1 – L13 in modified manuscript]:

"Our study focused on providing uncertainty estimations for many different interventions, not to provide an optimal intervention the given case study. Following this approach we made several simplifications, that must be resolved when an optimal intervention design is pursued. The linear interpolation between low intensity and high intensity interventions in figures 9 and 10 was used to obtain a first approximation of the uncertainty for interventions at different intensities. These figures are useful to illustrate how confidence levels and differences in relative uncertainty between interventions affect the expected effect but should not replace new calculations in intervention design – in part because our study provides only two support points, which is insufficient to either support or oppose the assumption of linearity. Rather, they may help to provide a starting point for a new set of calculations. Second, we marginalised the estimation uncertainty in figure 10 by averaging, to show how model uncertainty implicates intervention design. Practical use of exceedance levels should include estimation uncertainty of those levels --- which can be reduced by increasing the number model evaluations if necessary. Finally, the interventions studied in this paper are single archetypes, while in reality designs would likely be a combination of multiple archetypes (e.g. side channel combined with lowering of the surrounding floodplain). Our study does not provide support for claims regarding such compound interventions, which would require further study."

We further note that the variation alongside the river is no source of information for Figure 9, as this is strictly based on model prediction as x_m (the location of maximum effect). To

clarify this, as well as clarify the use of the error bars in this figure, we changed the caption of Figure 9 from:

"The expected effect (E(ΔH)) against the 90% confidence interval, (...)"

To

[caption of figure 9, p17 in modified manuscript]:

"The expected effect (E(ΔH)) at $x_m$ against the 90% confidence interval, (...) The error bars depict the estimation uncertainty (90\% interval). (..)"

[R 1.7] p17- exceedance probabilities If I understand correctly, this now linearly interpolates exceedance probabilities rather than relative uncertainty. It's not clear why this is a permissible inference. Are there characteristics of the distributions that allow this jump? What makes (all!) exceedance probabilities linear if relative uncertainty is constant with regard to expected effect? I assume at least some kind of symmetry comes into play? Even if the assumption of linearity doesn't end up holding completely, I would be in support of keeping this analysis in the paper, with some representation of the uncertainty in uncertainty involved (as for Figure 9). I think it's very important to be able to say something about the extremes of the distribution, not just about the size of the uncertainty.

[Reply to R1.7] Since the exceedance levels are directly related to the support points on the (cumulative) density function, the implicit assumption in the linear interpolation is that the transformation of the distribution function for low to high is linear with respect to the expected value. This does not necessarily mean that either distribution should be symmetrical, or even that the distributions share the same distribution. Given that we only have two points (low and high intensity), a linear transformation is the only model we can support. To better clarify the purpose of this figure, as well as the uncertainty involved in the estimation, we made the following modifications:

We wrote

[P17, L8-L10]:

"For example, figure 10 (a) shows various exceedance probabilities for /flpsmooth/, linearly interpolated between the low-intensity and high-intensity variants as function of the expected effect. "

We changed this to:

[P17, L2-L7 in modified manuscript]

"To illustrate how the exceedance probability can be used to guide intervention design, we linearly interpolated between the low-intensity and high-intensity variants as function of the expected effect (Figure 10). In this figure, the estimation uncertainty is not taken into account. Given that we only have two points (low and high intensity), a linear transformation is the only model we can support. Although we acknowledge that other models are possible, we included this as a first approximation, because it illustrates information available for decision makers."

We added the following to the caption of Figure 10:

[caption of figure 10, p19 in modified manuscript]

"Note that all lines show the expected value of the exceedance level; the estimation uncertainty is not depicted."

[R1.8] p19 Discussion It would be useful to mention that the study only considered individual
interventions, whereas it would be possible to combine measures in practice. I assume the combined effect is unlikely to be a linear combination of individual effects, so I don't think this study can be used to support any claims regarding such combinations. It would, however, be useful to highlight it as future work and speculate about possible issues that might crop up.

[Reply to R1.8] We agree with the reviewer that a combination of effects would not be linear and added the following section to the discussion (this follows our modification of P15, L25-L28 in response to R1.6):

[p20, L10-L13 in modified manuscript]

"Finally, the interventions studied in this paper are single archetypes, while in reality designs would likely be a combination of multiple archetypes (e.g. side channel combined with lowering of the surrounding floodplain). Our study does not provide support for claims regarding such compound interventions, which would require further study."

Technical comments

[R1.9] Table 1 The link with subsections in 2.3 could perhaps be emphasised, e.g. by using consistent ordering and grouping terms

[Reply to R1.9] In the manuscript, the grouping in table one was [Classification error, Main channel Roughness, Vegetation Height, Vegetation density, non-vegetation roughness], while the subsections in 2.3 were [main channel roughness, floodplain roughness, classification error matrix].

We now changed table 1 such that the order and grouping are identical to the subsections.

[R 1.10] p7 L14 "general extreme values distribution" Should be "Generalized Extreme Value
distribution"?

[Reply to R 1.10] That was indeed intended. We changed 'general extreme values distribution' to 'Generalized Extreme Value (GEV, Weibull variant) distribution'

[R 1.11] p7 L20 "a 95% confidence limits at 0.31 m and 1.0 m." drop the "a``

[Reply to R 1.11] We dropped the "a", such that the sentence now reads:

"(..) highly asymmetrical distribution with a mean of approximately 0.58 m and 95% confidence limits at 0.31 m and 1.0 m."

[R 1.12] p14 Table 2 Should "coefficient of variation" be average relative uncertainty E(Ur90)?
Otherwise, this raises the question of whether relative uncertainty was indeed needed?

[Reply to R 1.12] We corrected the caption to "Expected relative uncertainty along the river (average ± the standard deviation)"

[R 1.13] Figure 7 y axis label should read Ur90? (9 is missing). Figure 9 also seems to have some text rendering issues

[Reply to R 1.13] Unfortunately, the labels on the y-axis in both figures render correct in our PDF readers (showing Ur90, indeed); so we are not sure how to fix this issue. We will take this up if the manuscript is approved for final typesetting.

[R 1.14] p18 L17 "greatly the unexplained variance" Word missing (increased?)

[Reply R 1.14]

We wrote

[P18, L16-L17]:

"They observed that the removal of existing stochastic elements (e.g., a vegetation polygon) in exchange for new ones greatly the unexplained variance (...) "

We changed this to:

[P18, L17-L18 in modified manuscript]:

"They observed that the removal of existing stochastic elements (e.g., a vegetation polygon) in exchange for new ones greatly increased the unexplained variance (...) "

**Reply to referee 2**
We thank the reviewer for his time spend reviewing our manuscript and his suggestions for improvement. Below we have replied to the various comments made by the reviewer.

[R2.1] Overall the paper is well structured, well written and quite clear. However, I have a number of reservations regarding the manuscript. Several of these overlap with the points raised by Joseph in his comments, so I won't discuss those here in any further detail.

[Reply to R2.1] We thank the reviewer for his kind remarks. We acknowledge that some comments overlap with the first review and will refer to our reply to the first review where applicable.

[R2.2] My main reservation is regarding the contribution that the paper is trying to make. In the abstract the authors in the second sentence mention the role of model-based analyses in supporting decision making, while ending the abstract with claims about model-based decision making. In the middle of the abstract, the authors mention a new metric 'relative uncertainty'. Next, they argue that using this new metric they can provide insight into the uncertainty about the effects of a variety of flood risk reduction intervention.

I have two reservations here. First, regarding the claims related to decision making. In my view, these claims are not well developed in the paper. Outside of the introduction and conclusion, the term itself only appears once. Moreover, if I look at the presented results and their visualization, I doubt these would be used by a decision maker or even someone directly advising a decision maker. Rather, the presented analyses are useful for people working on the design of flood risks reduction strategies, while some of the results might indirectly be used in decision making. As such, I would suggest removing the term from the title as well as tone done any other claims made in the abstract and introduction. Alternatively, the authors would need to expand their discussion in the main text on how providing insights into the range of uncertainty about expected effects of measures can assist decision making on flood risk. This, however, requires discussing notions such as robustness (see e.g., McPhail et al., 2018) and flexibility, as well as discussions on well characterized uncertainty (i.e., you have a meaningful pdf) and Knightian or deep uncertainty (i.e., for whatever reason you don't have a meaningful and uncontested pdf).

[Reply to R2.2] We acknowledge that our use of the term "decision making" was intended to be narrower, and specific to decisions regarding the design of interventions. As the reviewer might well know, the established procedures for the design and approval of river interventions in The Netherlands (in Dutch, "Rivierkundig beoordelingskader") is highly model-based. It was inspired by this context that we intended to frame our work. We think it is to this the reviewer refers when he says "Rather, the presented analyses are useful for people working on the design of flood risks reduction strategies, while some of the results might indirectly be used in decision making."
So rather than being used directly in decision making, these models, and the extension offered by our work, would assist in decision making.

To avoid a too liberal use of the term 'decision making' in the context of the literature suggested by the reviewer, we made the following modifications.

The title was:
"Uncertainty quantification of flood mitigation predictions and implications for decision making."

We changed it to:
[P1 title in new manuscript]
"Uncertainty quantification of flood mitigation predictions and implications for interventions."

In the abstract we wrote:

"Finally, we show that for a defined standard of acceptable uncertainty, interventions need to be over-designed to meet this standard, and by how much. In general, we conclude that the uncertainty of model predictions is not large enough to invalidate model-based decision making, nor small enough to neglect altogether. Instead, uncertainty information can be used to improve intervention design and enrich the decision making process."

We changed this to:

[P1 L14-17 in modified manuscript]
"Finally, we show that for a defined standard of acceptable uncertainty, interventions need to be over-designed to meet this standard. In general, we conclude that the uncertainty of model predictions is not large enough to invalidate model-based intervention design, nor small enough to neglect altogether. Instead, uncertainty information is valuable in the selection of alternative interventions."

[R2.3] My second reservation is with the new metric itself. As also indicated by Joseph, this metric is closely related to the coefficient of variation. Moreover, the more theoretical discussion of this new metric is confined to one short paragraph around equation 5. Too play devils advocate: what is the merit of publishing a paper whose only contribution is a single equation closely related too an already established metric? The case study mainly serves to establish the value of this metric, while the models are taken from earlier work. If you insist on having the metric as a key contribution, than a comparative perspective would be more appropriate. So, what other metrics already exist that could serve a similar function? Classic robustness metrics as well as the coefficient of variation would be logical candidates. How is this metric different from these, and what are the relative merits of the new metric relative to the others?

[Reply to R2.3] We agree that the key contribution is not the relative uncertainty. In response to remark [R1.2] we have removed the focus on this metric.

[R2.4] Basically, both reservations have to do with how the authors currently position their work. I am having profound reservations regarding the first two paragraphs on page 3.

First, the authors cite Pinter on fuzzy math, suggesting that not providing explicit uncertainty quantification leaves decision makers free to interpret the uncertainty in any way they like. My argument would be that the converse also often happens. As for example elaborated nicely in Pilkey and Pilkey-Jarvis (2007), a lot of quantitative analysis for supporting decision making becomes useless arithmetic that is used strategically. In a different line of literature, researchers working on post normal science often claim that many models have not even one significant digit. That is, models give only a false sense of precisions. A third line of research, as exemplified by Sarewitz (2004) shows how more research and increasing efforts to quantify uncertainty can often make environmental controversies worse. In short, the relationship between models, model results, uncertainty about models results and decision making processes is quite a bit more complicated than the authors seem to think. It would benefit the manuscript it the claimed contribution is better positioned relative also to these strands of literature.

[Reply to R2.4] As motivated in [Reply to R2.2], we did not aim to focus on the decision-making process. The authors are aware of the post-normal science discussion, and concede that quantitative information can be used strategically, especially if there is too much focus on technical uncertainties to create superfluous knowledge which is not relevant to the policy process (see e.g. Warmink et al., 2017). We would not argue for the results of such uncertainty quantification (or model results in general) to take the centre stage in decision making, but merely to provide valuable information to assist in that process. We appreciate the suggestions of the reviewer to embed our work in the suggested literature, and have made the following modifications considering these suggestions and the intended scope of our work (cf. [Reply to R2.2]).

*Warmink et al. 2017: (P4592) in Water Resour Manage (2017) 31:4587–4600. DOI 10.1007/s11269-017-1767-6.*

The following paragraph is removed:

[p3 L3-10]
"To this date, there is little evidence in literature of explicit uncertainty quantification for model predictions of flood mitigation measures, or of other human intervention in river systems. Lack of explicit uncertainty quantification leaves room for what Pinter (2005) calls ``fuzzy math" --- free interpretation of model uncertainty --- by decision makers. In his example, building permits for floodplains were easily granted on the premise that individual effects are negligible compared to model uncertainty. Since the cumulative effect of all those permits is not negligible (Pinter et al., 2008), this interpretation likely overestimates the effect of uncertainty. An opposite example is given by Mosselman (2018), who reports that large, quantified uncertainties in absolute flood levels are sometimes ignored for intervention effects under the assumption that uncertainties `cancel out'.
This assumption is incorrect Berends et al. (2018), but further study is needed to determine how large the uncertainty actually is."

We added the following paragraph:

[P2 L9-21 in modified manuscript]

"Proper understanding and communication of uncertainty is important both for scientists and decision makers (Pappenberger and Beven, 2006; Uusitalo et al., 2015). Maier et al. (2016) distinguished three complementary paradigms for modelling to support decision making under (deep) uncertainty: (1) using the best available knowledge, (2) quantifying uncertainty and sensitivities of key parameters and (3) exploring multiple plausible outcomes. One advantage computational models can bring to decision making for river engineering is an assessment of the effect of the planned intervention on hydraulics, such as water levels, flow velocities and the morphodynamic response of the river bed. While hydraulic effects are not the only impact of interventions (see e.g. Straatsma et al., 2017, for effects on biodiversity) they are considered important: for the 39 interventions of the 2.3 billion 'Room for the River' program in The Netherlands, the hydraulic effect as predicted by models was a precondition for any design "to be taken seriously at all" (Klijn et al., 2013). Mosselman (2018) reported that quantified, large uncertainty in flood water levels is sometimes played down when assessing effects, under the assumption that systematic errors cancel out when subtracting the intervention case from the reference case. However, it was recently demonstrated in an idealized study that uncertainty in flood water levels does not cancel out but could be significant compared to the effect, and sensitive to the specific design of the intervention (Berends et al., 2018). Therefore, uncertainty quantification of effect studies for a real-world case study is needed to assess the implications for the design of interventions."

Furthermore, the rest of the introduction was revised to accommodate this new paragraph.

In the discussion, we wrote:

[p20, L17-21]
"In literature, there are several examples of probabilistic and risk-based approaches in designing flood adaptation strategies (Lendering et al., 2018; Klijn et al., 2018). Model uncertainty is generally not explicitly quantified as part of such approaches, but could be a natural addition to such approaches. This is especially so when model uncertainty influences intervention design (Hall and Solomatine, 2008; Beven et al., 2018). The quantification and visualisation of uncertainties here may be a step towards this goal."

We changed this to:

[p20, L27 – p21, L4 in modified manuscript]
"Models can play a role in decision support under (deep) uncertainty and the uncertainty quantification is an important step to model the future (Maier et al. 2016). However, too much focus on quantifiable uncertainty should be avoided. Warmink et al. (2017) documented several examples where too much focus on quantifiable (also "statistical" or "technical") uncertainty resulted in knowledge creation that did not contribute to the policy process ('superfluous knowledge'). A key idea is that other uncertainties exist that cannot be

reduced by more data or simply cannot be quantified. Examples of 'social uncertainty' include fundamental disagreement between experts, lack of trust in scientific results, ambiguity and diverging narratives based on the same empirical basis (Sarewitz 2004, Bruchnach, 2011, Warmink 2017). Coping with such uncertainties may require adaptive river management (Pahl-Wostl, 2009) and exploration of multiple alternate futures (The third paradigm of Maier et al. (2016)). Given these considerations, there are several probabilistic frameworks of flood risk adaptation strategies to which explicit quantification of model uncertainty is a natural and required extension. In this paper, we demonstrated that model predictions of human intervention in rivers are sensitive to uncertainty and can substantially influence design decisions. Therefore, we argue that trusting only on the best available knowledge and, by extension models, is not sufficient for planning interventions in river systems."

**Minor remarks**

[R 2.5] Might it not be more convenient to show the interventions (2.2.1-2.2.6) in a table?

[Reply to R 2.5] Our choice for subsections instead of a different format (e.g. a 'box') was mainly motivated by the limitation of the NHESS template rules. If approved for final typesetting, we will confer with the journal whether a different format is possible.

[R 2.6] What is the runtime of the detailed model for a single run?

[Reply to R 2.6] The runtime of a single model run was approximately 2.5 hours. We added this information to section 2.2.

[R 2.7] Page 18, line 17, "for new ones greatly the unexplained" I guess some words are missing here

[Reply to R 2.7] The missing word is "increased". For our answer we refer to our reply to [R 1.14]

[R 2.8] Page 17, line 4, partly overlapping with Joseph's comment, but what justifies the linear
interpolation?

[Reply to R 2.8] We refer to our answer to [R 1.7]

[revised manuscript text omitted]

---

## Referee Report (RR1)

**Review of Uncertainty quantification of flood mitigation predictions and implications for interventions (nhess-2018-325-m)**

In their study, the authors quantified the uncertainty of flood mitigation interventions on the Dutch River Waal for 39 different sources of uncertainty and 12 mitigation interventions.  The goal of this study was to quantify the uncertainty in the predicted changes in river levels associated with the application of the mitigation interventions.  To accomplish this goal, the authors developed a "relative uncertainty" parameter which they defined as the ratio between the confidence interval and the expected effect to assess the uncertainty in the predicted change in water levels for each of the mitigation interventions. The authors' found that uncertainty scales with discharge magnitude where there is greater uncertainty with higher discharge and less uncertainty with lower discharge conditions.  However, interventions with a similar magnitude of water-level reduction did not necessarily have the same level of uncertainty. The results showed that large-scale vegetation removal had much higher uncertainty than other mitigation interventions. The authors also argue that for a "defined standard of acceptable uncertainty", interventions need to be over-designed to ensure they meet their objective.

**General Comments:**

This paper provides an interesting case study that illustrates how uncertainty analysis might be used to inform project element selection in a comprehensive flood mitigation project. However, I believe the paper would be improved by briefly addressing the following issues.

1.  The authors should explain what they mean by "defined standard of acceptable uncertainty". It is the reviewers experience that "acceptable level of uncertainty" tends to be a very-grey area in hydraulic and other types of modeling.
2.  The analyses presented in this paper is essentially focused on the uncertainty associated with the selection of the roughness parameter in the hydraulic model.  Given this study's sole focus on this parameter, the analysis undertaken in this paper is an incomplete uncertainty analysis. The authors should discuss why they felt assessing just uncertainty associated in determining the roughness parameter was enough to adequately characterize the magnitude of uncertainty in the hydraulic modeling results.
3.  The authors should also discuss how much uncertainty analysis is enough to adequately inform decision making on interventions for flood-level reduction.

**Specific Comments:**

1.  Page 3 -Line 29 and Page 5 – Line 13: Spur "dams" or wing "dams" are a misnomer introduced by Pinter et al.  The official terms for these structures are spur dikes and wing dikes (see Parchure, T. M. 2005. Structural methods to reduce navigation channel shoaling. U.S. Army Corps of Engineer Research and Development, Center, Coastal and Hydraulic Laboratory, Vicksburg, Mississippi).
2.  Page 5 – Lines 9 and 17: I find the use of low-intensity and high-intensity confusing in this context.  Are you talking about discharge magnitude or flow duration?  Please clarify.
3.  Page 9 – line 20:  There are an extra ")." After Stefanou, 2009)
4.  Page 18 – lines 10 -13:  This is more of a comment than a suggestion.  It would have been interesting to assess the uncertainty on the cost benefit analyses undertaken for the flood mitigation interventions.  Showing the potential impact of the hydraulic modeling uncertainty

on the benefits of the mitigation interventions would be how to poignantly illustrate the authors point here to decision makers.

5.  Page 18 – line 29 – Please remove the comma after evaluations

---

## Author Response (AR2)

**Combined point-by-point reply to the comments**

Iteration: minor revision

All referee comments are numbered by referee number and comment number, e.g. [R1.2] for the second comment of the first referee. The author's response is colour coded:

Black - Reviewer comment
Blue - Author reply
Green - Proposed change in manuscript

**Replies to referee 1**

On the whole, the authors' responses adequately address my concerns, and I support publication of this article. I would, however, like to comment on one of my criticisms. I had originally suggested that a major contribution was in approximating uncertainty, but that 1) some extra support was needed for the argument that linear interpolation was an appropriate approximation, and 2) the paper would benefit from additional discussion of the uncertainty in the approximation. Rather than providing these additions, the authors have weakened their argument, emphasising that this linear interpolation is only a first approximation that cannot (yet) be used as a technical tool for obtaining uncertainty estimates (...) I find this to be an adequate solution, as I was particularly impressed during the first review that this uncertainty approximation method "made the results of the analysis more accessible to a broader audience." The authors have played to their strength in emphasising the conceptual rather than technical use of their results.

Nevertheless, there are a couple of points where I think the selected wording could better reflect the authors's decision in this regard:

We thank the referee for taking the time to review our revised manuscript, his appreciation for our reply to his earlier comments and his additional suggestions in this iteration.

[R1.1] p1 L7 "establish relationships to aid in decision making"
-> "establish approximate relationships to aid in decision making" or "investigate relationships that may aid decision making"

We modified this line from:

"Our objective is to investigate the uncertainty of model predictions of intervention effect and to establish relationships to aid in decision making."

To:

"Our objective is to investigate the uncertainty of model predictions of intervention effect and to explore relationships that may aid in decision making."

[R 1.2] p3 L11 "Our second objective is to establish a relationship"
-> "establish an approximate relationship" or "explore the relationship"

We modified this line from:

"Our second objective is to establish a relationship between the expected reduction of flood levels and the uncertainty, to aid in the decision making process."

To:

"Our second objective is to explore the relationship between the expected reduction of flood levels and the uncertainty, to aid in the decision making process."

[R 1.3] p21 L25 "The second objective was to establish a relationship between the expected effect and its uncertainty, to aid in the decision making process."
-> "establish an approximate relationship" or "explore the relationship"

We modified this line from:

"The second objective was to establish a relationship between the expected effect and its uncertainty, to aid in the decision making process."

To:

"The second objective was to explore the relationship between the expected effect and its uncertainty, to aid in the decision making process."

[R 1.4] p1 L10 we show that intervention effect uncertainty behaves like a traditional backwater curve with a constant relative uncertainty value
-> "behaves approximately like" or "with an approximately constant"

We modified this line from:

"Using this metric, we show that intervention effect uncertainty behaves like a traditional backwater curve with a constant relative uncertainty value."

To:

"Using this metric, we show that intervention effect uncertainty behaves like a traditional backwater curve with an approximately constant relative uncertainty value."

[R 1.5] The paper could also benefit from a final proof-read. Here are a couple of typos/errors I noticed:

p2 L16 "2.3 billion" -> euro?
p4 L15 "gradually increasing resolution" -> increasing grid size or decreasing resolution?
p5 L9 "Each interventions" -> each intervention
p6 L21 "as decreases vegetation friction" -> and decreases...
p8 L1 "are designed" -> are designated
p10 L6 "is that a" -> is a
p14 "The novelty of Fig. 6 these results" -> "The novelty of the Fig. 6 results"
p14 "coral estimation" -> CORAL
p20 L2 "intervention the given case study." -> for the given

All above suggestions have been adopted in the revised manuscript, as well as other small changes after a further proof-read.

**Replies to referee 2**

The authors have made a substantial effort to address my comments and concerns. The modifications and additions largely have addressed my initial concerns. Given the updated manuscript, I still have a few questions and suggestions that might be of use in further developing the paper.

We are grateful to the referee for taking the time to review our revised manuscript and for suggesting additional changes.

[R 2.1] First, you introduce relative uncertainty as a useful metric. I still think that the manuscript could benefit from a broader discussion and preferably a comparison with other metrics. For example, relative uncertainty is quite close to a signal to noise ratio. On theoretical grounds, why would you prefer one over the other, or empirically do they produce different insights for this specific case? Alternatively, perhaps look at (McPhail, Maier et al. 2018) and position relative uncertainty in light of the more conceptual description of robustness metrics provided there.

McPhail, C., H. R. Maier, J. H. Kwakkel, E. Giuliani, A. Castelletti and S. Westra (2018). *"Robustness metrics: How are they calculated, when should they be used and why do they give different results*?" Earth's Future.

When introducing the relative uncertainty($U_R$), we state [p11 L31]: *"(..) For this, we use an adapted version of the coefficient of variation, defined as: (..)".* The referee notes that the relative uncertainty metric is closely related to the signal-to-noise ratio (SNR). This is correct, as the SNR is defined as the reciprocal of the coefficient of variation (CV), and if the distribution of the effect is normal, CV and $U_R$ are proportional. Our consideration for using $U_R$ over CV is interpretational clarity, as the size of the uncertainty (measured by the size of the confidence interval) is expressed as a percentage of the expected effect. Using the CV or

SNR would produce no different insights. To better position $U_R$ we modified the following line [p12 L1]:

"The $U_{R;90}$ is expressed in percentages. Therefore, a $U_{r;90}$ of 100% means that the 90% MCI is as large as the average effect"

To:

"We use $U_{r;90}$ over the coefficient of variation (which is defined as the standard deviation on the expected value) to directly express the size of the largest used confidence interval in percentages of the expected effect. Therefore, a $U_{r;90}$ of 100% means that the 90% MCI is as large as the average effect."

[R 2.2] I struggle with the interpretation given by the authors of figure 8. You write that MINEMBLOW and FLPSMOOTH are most uncertain. I guess this has to do with the bandwidth of outcomes covered by the densities in figure 8. However, based on visual comparison, the difference between MINEMBLOW and GROYNELOW seems to be quite marginal. So how did you arrive at this statement and might it be possible to annotate figure 8 in such a way that it becomes easier for the reader to come to the same conclusion?

The referee states that the differences between MINEMBLOW and GROYNELOW are marginal when compared visually. We stated [p16 L3]:

"In general we observe that (i) flpsmooth and minemblow are clearly more uncertain than the other four interventions and (ii) that (...)"

Indeed, we did not specify how we arrived at that conclusion. Therefore we modified this line to the following:

"In general we observe that flpsmooth and minemblow are more uncertain than the other four interventions. This can be seen by their higher values for $U_{r;90}$, especially for the high-intensity variants. We also observe that (...)"

Minor comments

[R 2.3] In the abstract, you write "affects predictions of flood mitigation strategies", I suggest rephrasing this to "affects predictions of the effects of flood mitigation strategies"

We modified this line from:

"However, the predictions of computer models are inherently uncertain, and it is currently unknown to what extent that uncertainty affects predictions of flood mitigation strategies."

To:

"However, the predictions of computer models are inherently uncertain, and it is currently unknown to what extent that uncertainty affects predictions of the effects of flood mitigation strategies."

[R 2.4] on page 2 you claim that more detailed models are needed given the increasingly complicated nature of intervention designs. I am not sure whether this actually follows.

The referee refers to the following sentence:

"First, there is a clear practical need for detailed models given the increasingly complicated design of interventions, which moves from traditional flood prevention (building dikes, or levees) to more holistic designs."

What we mean by this is that the detail of the model and the detail of the intervention design should agree. For example, consider an intervention to remove a forested plot of land from an inner bend floodplain for the water to shortcut the river bend during overbank discharge. This would ideally not be solved using a one-dimensional routing model, but a two-dimensional model with a sufficiently small grid size and geographical detail would probably suffice. To stress this we changed the sentence from:

"First, there is a clear practical need for detailed models given the increasingly complicated design of interventions, which moves from traditional flood prevention (building dikes, or levees) to more holistic designs."

To

"First, there is a practical need for sufficiently detailed models given the increasingly complicated design of interventions, which moves from traditional flood prevention (building dikes, or levees) to more holistic designs."

[R 2.5] On page five, line 10 you touch on the problem of implementation uncertainty. Realizing that it is outside scope for this paper, I am still curious whether you have any thougths on how to bring it into this kind of analysis. For example, you could include an additional source of uncertainty with some variation on how something is implemented. That would provide you with insight into the sensitivity of your designs to this implementation uncertainty.

Implementation uncertainty would indeed be an interesting extension. The technical implementation of this is straightforwardly achieved by adding additional sources of uncertainty. For example, stochastic groyne heights to express uncertainty of successful groyne lowering. However, to our knowledge there is no literature to base the input distributions on. This would require a comprehensive study on differences between designed measured and implemented measures. We added this on page 5:

"In reality, there may be a discrepancy between the *as designed* state and the actual ('*as built*') state. In theory, this discrepancy could be considered as an additional source of uncertainty. However, because there is no literature to support any assumptions regarding such discrepancy, we do not take this into account here."

[R 2.6] The focus of the paper is on the uncertainty in the estimated effects, which during design is quite relevant. However, what also matters is whether the ranking of solutions changes due to this uncertainty.

While we agree with the referee, we purposefully don't rank alternatives in this paper. This would have been proper if a formal objective was set with regard the hydraulic effect to be obtained at this location. This was not the case, and we did not intend to present one alternative as better than another. We also note, that hydraulic effect is only indicator by which to rank alternatives. To discuss this, and to point to other papers that go in further depth regarding this, we added the following to the discussion

"Finally, our study was limited to the effect of interventions with respect to water level lowering. We did not consider other indicators, such as economic, ecological or societal costs and benefits, as was done by Straatsma *et al.* (2019) (albeit without addressing uncertainty). Multidisciplinary and multisectoral assessment of interventions including uncertainty is recommended to determine trade-offs in river management. Coral provides a computationally effective method to do so."

[R 2.7] P14, line 9, coral should probably be capitalized.

We now consistently refer to the method as CORAL. (small caps)

**Replies to referee 3**

In their study, the authors quantified the uncertainty of flood mitigation interventions on the Dutch River Waal for 39 different sources of uncertainty and 12 mitigation interventions. The goal of this study was to quantify the uncertainty in the predicted changes in river levels associated with the application of the mitigation interventions. To accomplish this goal, the authors developed a "relative uncertainty" parameter which they defined as the ratio between the confidence interval and the expected effect to assess the uncertainty in the predicted change in water levels for each of the mitigation interventions. The authors' found that uncertainty scales with discharge magnitude where there is greater uncertainty with higher discharge and less uncertainty with lower discharge conditions. However, interventions with a similar magnitude of water-level reduction did not necessarily have the same level of uncertainty. The results showed that large-scale vegetation removal had much higher uncertainty than other mitigation interventions. The authors also argue that for a "defined standard of acceptable uncertainty", interventions need to be over-designed to ensure they meet their objective.

General Comments:

This paper provides an interesting case study that illustrates how uncertainty analysis might be used to inform project element selection in a comprehensive flood mitigation project. However, I believe the paper would be improved by briefly addressing the following issues.

We thank the reviewer for taking the time to review our manuscript and to suggest changes for improvement.

[R 3.1] The authors should explain what they mean by "defined standard of acceptable uncertainty". It is the reviewers experience that "acceptable level of uncertainty" tends to be a very-grey area in hydraulic and other types of modeling.

The referee cites a sentence in the abstract:

[p1 L15]: "Finally, we show that for a defined standard of acceptable uncertainty, interventions need to be over-designed to meet this standard"

This sentence references the following part in the paper:

[p16 L11]: "For decision making, it may be useful to estimate how much uncertainty is expected for a given expected effect. When considering different flood mitigation measures, the choice may depend on the level of acceptable uncertainty."

We then go on to show two ways in which such level can be defined. First, by the size of the confidence interval and second by a given exceedance probability. The referee suggest that we explain what we mean by a "defined standard of acceptable uncertainty". In this paper, we give two ways by which this standard can be set by decision makers. We do not set this standard ourselves, nor would we aim to. To better reflect our intent in the paper we modified the following sentence in the abstract [p1 L15]:

"Finally, we show that for a defined standard of acceptable uncertainty, interventions need to be over-designed to meet this standard."

To

"Finally, we show how a level of acceptable uncertainty can be defined and how this can affect the design of interventions."

[R 3.2] The analyses presented in this paper is essentially focused on the uncertainty associated with the selection of the roughness parameter in the hydraulic model. Given this study's sole focus on this parameter, the analysis undertaken in this paper is an incomplete uncertainty analysis. The authors should discuss why they felt assessing just uncertainty

associated in determining the roughness parameter was enough to adequately characterize the magnitude of uncertainty in the hydraulic modeling results.

We discuss our selection of uncertainty sources in paragraph 2.3:

[p7 L1]
"Sources of uncertainty for this river were identified by Warmink *et al.* (2011b) using expert elicitation. The main sources were (a) boundary conditions and (b) main channel friction and to lesser extent (c) friction by vegetation, (d) geometry and (e) weir and groynes formulations. In this study we follow the design approach taken in the Room for the River program, which assumes the boundary conditions given and stationary for a certain design return rate (which is $T_{1250}$) and therefore not a source of uncertainty. Following Warmink et al. (2013b), we take into account uncertainty in the main channel and classification errors in the land-use maps. Additionally, we take into account uncertainty in vegetation parameters. This leads to 39 stochastic parameters (see table 1)"

To better reflect that our choice of uncertainty sources if motivated on earlier literature, we modify this section as follows:

"Sources of uncertainty for this river were identified by Warmink *et al.* (2011b) using expert elicitation. The main sources were (a) boundary conditions and (b) main channel friction and to lesser extent (c) friction by vegetation, (d) geometry and (e) weir and groynes formulations. In this study we follow the design approach taken in the Room for the River program, which assumes the boundary conditions given and stationary for a certain design return rate (which is $T_{1250}$) and therefore not a source of uncertainty. Warmink et al. (2013b) considered the hydraulic roughness parameters to be the most important parameter based on literature. We adopt this assumption and take into account uncertainty of the main channel friction, vegetation parameters and classification errors in the land-use maps. The total number of stochastic variables is 39 (see table 1)"

[R 3.3] The authors should also discuss how much uncertainty analysis is enough to adequately inform decision making on interventions for flood-level reduction.

The referee raises an interesting concern, namely to what extent model projects can inform decision making. We addressed to some extent in our discussion on the assumptions made in an uncertainty analysis [p20 L21-35] and secondly on the role of models in decision-making frameworks [p21 L1-14]. A general and more in-depth discussion on how uncertainty can inform decision making is given by Maier *et al.* (2016), to which we refer in our manuscript.

Specific Comments:

[R 3.4] Page 3 -Line 29 and Page 5 – Line 13: Spur "dams" or wing "dams" are a misnomer introduced by Pinter et al. The official terms for these structures are spur dikes and wing

dikes (see Parchure, T. M. 2005. Structural methods to reduce navigation channel shoaling. U.S. Army Corps of Engineer Research and Development, Center, Coastal and Hydraulic Laboratory, Vicksburg, Mississippi).

We thank the reviewer for correcting our reference to the American naming convention. We have changed 'dams' to 'dikes' accordingly.

[R 3.5] Page 5 – Lines 9 and 17: I find the use of low-intensity and high-intensity confusing in this context. Are you talking about discharge magnitude or flow duration? Please clarify.

It is neither discharge magnitude nor duration. The terminology was adopted from Straatsma & Kleinhans and refers to the magnitude of the intervention. To better show this we changed the following sentence on p5:

"Each intervention was implemented in a low-intensity and a high-intensity variant."

To

"Each intervention was implemented in a low-intensity and a high-intensity variant. Here, intensity refers to the magnitude of the intervention, for example a small reduction in groyne height (low-intensity) or a large reduction in groyne height (high-intensity)."

[R 3.6] Page 9 – line 20: There are an extra ")." After Stefanou, 2009)
We removed the superfluous " )."

[R 3.7] Page 18 – lines 10 -13: This is more of a comment than a suggestion. It would have been interesting to assess the uncertainty on the cost benefit analyses undertaken for the flood mitigation interventions. Showing the potential impact of the hydraulic modeling uncertainty on the benefits of the mitigation interventions would be how to poignantly illustrate the authors point here to decision makers.

This is an interesting point, and it touches on the comment of the second referee regarding ranking alternatives and the effect uncertainty might have on those rankings [R 2.6]. In this paper, we show how uncertainty may affect interventions based only on the hydraulic effect. We do not consider other indicators, such as economic, ecological or societal costs and benefits, as was done by other authors (e.g. Straatsma *et al.*, 2019, albeit without addressing uncertainty). A comprehensive uncertainty study including costs and benefits (and the uncertainties associated with those other indicators) would be interesting and challenging but is well beyond the scope of this current paper. We have added the following sections to the discussion to address this concern:

[revised manuscript text omitted]